# Causal Direct Preference Optimization for Distributionally Robust Generative Recommendation

**Chu Zhao**[1]  **Enneng Yang**[2]  **Jianzhe Zhao**[1]  **Guibing Guo**[1]

## Abstract

Direct Preference Optimization (DPO) guides large language models (LLMs) to generate recommendations aligned with user historical behavior distributions by minimizing preference alignment loss. However, our systematic empirical research and theoretical analysis reveal that DPO tends to amplify spurious correlations caused by environmental confounders during the alignment process, significantly undermining the generalization capability of LLM-based generative recommendation methods in out-of-distribution (OOD) scenarios. To mitigate this issue, we propose CausalDPO, an extension of DPO that incorporates a causal invariance learning mechanism. This method introduces a backdoor adjustment strategy during the preference alignment phase to eliminate interference from environmental confounders, explicitly models the latent environmental distribution using a soft clustering approach, and enhances robust consistency across diverse environments through invariance constraints. Theoretical analysis demonstrates that CausalDPO can effectively capture users' stable preference structures across multiple environments, thereby improving the OOD generalization performance of LLM-based recommendation models. We conduct extensive experiments under four representative distribution shift settings to validate the effectiveness of CausalDPO, achieving an average performance improvement of 17.17% across four evaluation metrics. Our implementation code is available at https://github.com/user683/CausalDPO.

## 1. Introduction

Recent advances in large language models (LLMs) have shown strong performance across diverse tasks (Chang et al., 2024; Liang et al., 2024), motivating their adoption in recommender systems (Liang et al., 2024; Wu et al., 2024c). Existing LLM-based recommenders typically (i) use LLMs to generate or refine user/item representations and embeddings (Zhao et al., 2025b; Ren et al., 2024), (ii) prompt LLMs with user histories or collaborative signals for next-item prediction (Zhang et al., 2025; Zhu et al., 2024), or (iii) distill LLM knowledge into lightweight models for efficient serving (Cui et al., 2024; Li et al., 2023a). These methods improve fine-grained preference modeling and enable reasoning-aware recommendation. To support generation-based recommendation, many works inject domain knowledge via supervised fine-tuning (SFT) (Dong et al., 2023). More recently, Direct Preference Optimization (DPO) (Rafailov et al., 2023) further aligns LLM outputs with user preferences by training on offline triples ¡context, positive item, negative item¿, encouraging the model to learn preference orderings and produce more personalized recommendations.

While these methods have achieved promising results, our empirical studies and theoretical analyses reveal critical limitations. Data used during SFT often contains environment-specific confounders, which can induce LLMs to learn spurious correlations dependent on such factors. Worse still, the DPO objective tends to amplify these spurious dependencies during the preference alignment process, thereby hindering out-of-distribution (OOD) generalization. In this paper, **environmental confounders refer to unobserved factors introduced by specific contexts or external conditions that underlie the training data**. Such factors can induce spurious correlations that deviate from the true causal mechanism, thereby weakening generalization on OOD datasets. For example, during COVID-19 lockdowns, demand for medical, fitness, and entertainment products rose simultaneously; A model might then mistakenly associate users' preferences for fitness or electronic goods with medical supplies. In this setting, COVID-19 is a prototypical environmental confounder. Other common sources include policy changes, social events or campaigns, platform popularity bias, and temporal/seasonal drift. As shown in Figure 1 (left), we

[1]Northeastern University, Shenyang, China [2]Shenzhen Campus of Sun Yat-sen University, China. Correspondence to: Guibing Guo and Jianzhe Zhao <guogb, zhaojz@swc.neu.edu.cn>.

*Proceedings of the 43rd International Conference on Machine Learning*, Seoul, South Korea. PMLR 306, 2026. Copyright 2026 by the author(s).

*Figure 1.* **Left**: This figure presents the number of interactions (frequency) for DPO-based models across item popularity groups, with popularity decreasing from G1 (head) to G5 (tail). **Middle**: This figure illustrates how LLM-based recommendation models can learn and amplify spurious correlations during preference alignment, and how the DPO mechanism further reinforces these spurious correlations. **Right**: Using a Structural Causal Model (SCM), this figure analyzes how environmental confounders $E$ affect the model and demonstrates how their influence can be mitigated via the backdoor adjustment criterion.

observe that after DPO training, the model significantly increases the interaction counts for high-popularity groups (e.g., G1–G2), while further reducing the interaction counts for long-tail groups (e.g., G4–G5). This amplifies the bias induced by popularity as a confounding factor, making the model more prone to rely on environment-driven spurious correlations rather than genuine preference signals.

Recent works (Jiang et al., 2024; Bao et al., 2024; Gao et al., 2024; Sakib & Das, 2024; Dai et al., 2024) have explored mitigation strategies for LLM-based recommendation under distribution shifts. For example, RW (Jiang et al., 2024) adopts a three-stage approach combining de-biased sampling, fairness-based reweighting, and re-ranking; $D^3$ (Bao et al., 2024) addresses score bias and content homogeneity via ghost token normalization and low-frequency token promotion; and SPRec (Gao et al., 2024) employs iterative adversarial training to suppress generation biases. However, these methods often target specific distribution shifts, while real-world recommendation data typically involves multiple, interrelated complex shifts, highlighting the need for a unified and flexible approach that generalizes robustly across various real-world distribution changes and scenarios.

In this work, we propose CausalDPO, a novel extension of DPO that incorporates causal invariance constraints to mitigate the impact of environmental confounders and reduce reliance on spurious correlations, thereby improving cross-environment generalization. We first construct a causal structural model to analyze how LLMs, during the SFT stage, can internalize spurious patterns caused by latent environment factors. We further show both empirically and theoretically that the DPO process exacerbates this issue by reinforcing such spurious correlations. To address this, we introduce a backdoor adjustment strategy that leverages soft clustering to group samples, implicitly modeling environmental factors without requiring explicit environment labels. We then impose a cross-group invariance regularization within the DPO objective to encourage consistent preference modeling across inferred environments. This fosters robust, causally invariant preference learning.

**Our contributions can be summarized as follows**: **(1)** We conduct the empirical study and theoretical analyses to demonstrate the risk of spurious correlation amplification in LLM-based recommendation systems, caused by environmental confounders during DPO-based preference alignment. **(2)** We introduce **CausalDPO**, a causality-aware enhancement of DPO that leverages soft clustering and backdoor adjustment, together with invariance regularization, to implicitly model and counteract environmental confounding effects. **(3)** Extensive experiments across four representative distribution shift scenarios demonstrate that CausalDPO significantly enhances the generalization ability of LLM-based generative recommenders under complex environments.

## 2. Preliminary

This section introduces two dominant LLM-based generative recommendation paradigms: *discriminative* and *direct generation*, and outlines their core optimization pipelines, particularly direct preference optimization.

**Task Formulation**. Given a user's interaction history $\mathcal{S}_u = \{i_h\}_{h=1}^N$, an LLM-based recommender $\mathcal{M}_\theta$ operates under one of two paradigms: (1) *Discriminative*: Select the most relevant item $i_p$ from a candidate set $\mathcal{C}$; (2) *Generative*: Directly generate a target item $i_g$ conditioned on a prompt $\mathbf{p}$, typically constructed as $\mathbf{p} = \text{Concat}[\mathcal{S}_u, \mathcal{P}_u, \mathcal{P}_i]$. Where $\mathcal{P}_u$ is the user profile and $\mathcal{P}_i$ is the item profile. Both paradigms are unified as: $i^* = \arg\max_{i \in \mathcal{C} \cup \mathcal{V}} P_{\mathcal{M}_\theta}(i|\mathbf{p})$, where $\mathcal{V}$ denotes the model's vocabulary space.

**Direct Preference Optimization (DPO)**. To align LLM outputs with user preferences without explicit reward modeling, the DPO objective is formulated as follows:

$$\mathcal{L}_{\text{DPO}}(\pi_\theta; \pi_{\text{ref}}) = -\mathbb{E}_{(x, y_w, y_l) \sim \mathcal{O}}$$
$$\left[ \log \sigma \left( \beta \log \frac{\pi_\theta(y_w|x)}{\pi_{\text{ref}}(y_w|x)} - \beta \log \frac{\pi_\theta(y_l|x)}{\pi_{\text{ref}}(y_l|x)} \right) \right], \quad (1)$$

where $y_w$ and $y_l$ are preference-aligned positive and negative samples (e.g., clicked v.s. non-clicked items), $\pi_{\text{ref}}$ is a frozen reference model, and $\beta$ controls policy deviation. DPO enables efficient, contrastive alignment of generative

recommendations with user behavior. Our empirical study reveals that using DPO to align with human preferences can amplify the influence of confounders and degrade performance. To address this, we propose a Causal Direct Preference Optimization method, which is a variant of DPO.

## 3. Methodology

This section first analyses how Direct Preference Optimization (DPO) tends to amplify spurious correlations caused by environmental confounders during the preference alignment process. We formulate a causal intervention objective to mitigate this issue and introduce a practical implementation based on soft clustering and invariance regularization. Through rigorous theoretical proof, we demonstrate that optimizing CausalDPO can encourage LLM-based methods to learn stable user preferences across environments, further enhancing the recommendation capability out-of-distribution.

### 3.1. Motivation: Spurious Correlation Amplification in DPO

As discussed in Section 2, LLM-based generative recommendation models are typically adapted to downstream tasks via SFT and DPO. However, data used during SFT and DPO often contains implicit environment-specific confounders (e.g., time, popularity, exposure bias). These confounders influence both the input features and the preference labels, inducing a spurious dependency structure. To better illustrate this issue, we further construct a Structural Causal Model (SCM) to characterize the relationship between the input text $x$ and the predicted target label $y_i$, aiming to uncover the underlying dependency structure and potential sources of bias during training. The dependencies among the edges in Figure 1, right part (a), are described as follows: (1) $E \rightarrow X$: This dependency indicates that the latent environment variable $E$ affects the distribution of input data $X$, i.e., $p(X|E)$, reflecting the environmental influence on data generation. (2) $X \rightarrow Y$: This represents the model's predictive process $p(Y|X)$, where the relationship between $X$ and $Y$ becomes deterministic when the model parameters $\theta$ are fixed. (3) $E \rightarrow Y$: This dependency captures the indirect influence of environment $E$ on the model's output $Y$ through the supervised fine-tuning process of LLMs, as $E$ affects how the model is fine-tuning. Since the environment $E$ affects the data distribution through the joint probability $p(X, Y|E) = p(X|E)p(Y|X, E)$. We further have:

$$\theta^* = \arg \min_{\theta} \mathbb{E}_{e \sim p_{tr}(E), (X,Y) \sim p(X,Y|E=e)}$$
$$[-\log \mathcal{M}_{\theta}(Y|X)], \quad (2)$$

where $p_{tr}$ denotes the training data distribution. From Figure 1 right part (a) and the above analysis, we can see that when directly optimizing the likelihood function, a statistical

dependency exists between the environment $E$ and the label $Y$ (i.e., $p(Y|E) \neq p(Y)$). As a result, the model implicitly learns a spurious causal path from $E \rightarrow Y$ during loss minimization—capturing correlations induced by specific environments $e$ in the training data. This spurious correlation is highly sensitive to distributional changes. In particular, when the test environment $E' \neq E$, the spurious correlation $p_{tr}(Y|E)$ learned during training no longer aligns with the true distribution $p_{ts}(Y|E')$, causing the model to rely on environment-specific noisy features that become invalid. Under distribution shift, the performance of the model $\mathcal{M}_{\theta}$ trained in Eq. (2) can be formalized as:

$$\text{Error}_{ts} \approx \mathbb{E}_{E'} \left[ \text{KL} \left( p(Y|X, E') \| \mathcal{M}_{\theta}(Y|X) \right) \right], \quad (3)$$

where KL($\cdot$) denotes the Kullback–Leibler divergence. When $\mathcal{M}_{\theta}(Y|X)$ overly depends on non-causal features associated with the environment $E$, the KL divergence increases significantly, severely impairing the model's generalization performance in unseen environments. Building on the above analysis, we directly present the following proposition (the proof provided in Appendix A.1).

**Proposition 3.1** (DPO amplifies spurious correlations and hinders generalization capabilities)**.** *In the context of LLMs, when the environmental confounder $E$ in the training data satisfies the following preference bias:*

$$p_{train}(E \mid y_w) > p_{train}(E \mid y_l), \quad (4)$$

*the environmental feature $E$ is more likely to appear in the preferred outputs $y_w$, DPO tends to learn spurious correlations between $E$ and the preferred output $y_w$. During the maximum likelihood optimization of the policy $\pi_{\theta}$ on preference pairs, such spurious preference signals are further amplified as the weight $w_E$ associated with the environmental feature function $f_E(y, E)$ increases iteratively during training. This yields a generalization bound under distribution shift, showing that the generalization error is upper-bounded by the mismatch between the training and test conditional distributions of $E$:*

$$\text{GenErr}(\theta) \leq 2C|w_E| \cdot \mathbb{E}_{p_{test}(x)} \left[ \text{TV}(p_{train}(E|x), p_{test}(E|x)) \right]. \quad (5)$$

*Where TV denotes the Total Variation Distance and $C$ is the constant. It is easier to judge $y_w \succ y_l$ under a specific environmental feature $E = e$, then the learned policy $\pi_{\theta}$ will over-rely on the environmental feature $E$ for preference distinction. This leads to significant performance degradation in the test environment when the distribution of $E$ is no longer consistent.*

### 3.2. Causal Objective: Backdoor Adjustment via Intervention

The above theoretical analysis suggests that improving the generalization of DPO relies on removing the influence of

confounders and guiding large language models to capture the stable preference patterns of users. Specifically, this can be achieved by optimizing the causal objective $(Y|do(X))$, which enhances the alignment with true preference signals. The $do$-operator represents an intervention on the input variable $X$, severing its dependence on potential confounders $E$ in the causal graph. This intervention removes the effect of $E$ on $X$, preventing the model from learning spurious correlations and enabling more robust causal modeling of user preferences.

When applying causal intervention methods to eliminate the influence of confounding factors, practical resource constraints often make it infeasible to estimate $p(Y|do(X))$ through actual physical interventions on the input variable $X$. For example, conducting randomized controlled trials or recollecting large-scale data under all possible environmental contexts to achieve complete environment randomization is prohibitively costly and difficult to implement in real-world applications. Therefore, a more practical approach is to learn from observational data. Specifically, to estimate the true causal effect of $X$ on $Y$, it is essential to block all backdoor paths induced by potential confounders. As shown in Figure 1 right part (b), by implementing the $do(X)$ intervention in the causal graph, we can sever the edge $E \rightarrow X$, thereby eliminating the backdoor path induced by $E$ and enabling causal preference modeling. Building on this principle, we have:

$$p(Y|do(X)) = \sum p_\theta(\hat{Y} \mid X, E = e) \cdot p(E = e)$$
$$= \mathbb{E}_{p_\theta(E)} \left[ p(Y|E, X) \right], \quad (6)$$

where $p_\theta$ denotes the prior distribution over environments, capturing the likelihood of different environments $E$ without observational data. The detailed derivation of the above formulas can be found in Appendix A.2. In real-world data, the environment $E$ is typically unobservable, making it challenging to explicitly characterize environmental factors and instantiate $p(Y|X, E)$. Moreover, existing studies(Wu et al., 2024c; Ding et al., 2025) have shown that even when explicit environmental information is available in the data, it may still be insufficient to effectively guide the model toward learning stable relationships that are robust to distribution shifts. To mitigate the confounding effects caused by the unobserved environment $E$, we propose a data-driven approach that employs **soft clustering** to implicitly learn environmental partitions from the data, thereby modeling and adjusting for the latent environment.

### 3.3. Architecture of CausalDPO

The core idea of CausalDPO is to infer pseudo-environment labels embedded in the output data via causality-driven soft clustering. These inferred labels are used to perform invariant policy learning conditioned by the environment,

enhancing the model's generalizability under distribution shift.

**Modeling Unobserved Environments via Soft Clustering**. Given a batch of input samples $X = \{x, y_w, y_l\}_{i=1}^{\mathcal{B}}$, where each $x$ contains user-specific contextual information and $\mathcal{B}$ is the batch size, we first obtain the corresponding hidden representations $h_i$ using the LLMs. These hidden states are then assigned to causal representations $z_i$ using a causal feature extractor $g(\cdot)$ to capture stable causal relationships between input and latent environment. The $g(\cdot)$ is defined as $g(h_i) = W_g h_i + b_g = z_i$, where $W_g \in \mathbb{R}^{d' \times d}$, and $d'$ is typically set to a lower dimension for clustering stability. After obtaining the causal representations $Z = \{z_i\}_{i=1}^{\mathcal{B}}$, we perform unsupervised environment discovery by first applying DBSCAN (Khan et al., 2014) for an initial hard clustering. We choose DBSCAN for its robustness to noise and its ability to capture arbitrarily shaped clusters, unlike K-Means which assumes spherical structure. We then compute the center $c_k$ for each discovered cluster as:

$$c_k = \frac{1}{|S_k|} \sum_{i \in S_k} z_i, \quad S_k = \{i \mid y_i = k\}, \quad (7)$$

where $y_i$ denotes the cluster assignment of sample $i$, and $S_k$ is the index set of samples assigned to cluster $k$. To derive a soft clustering assignment, we measure the distance between each representation $z_i$ and cluster center $c_k$ using Euclidean distance $D_{ik} = \|z_i - c_k\|_2$. We then transform the distances into probabilities via the softmax function:

$$p_{ik} = \frac{\exp(-D_{ik})}{\sum_j \exp(-D_{ij})}, \quad p_i \in \Delta_K, \quad (8)$$

where $p_{ik}$ denotes the probability that sample $i$ belongs to cluster $k$, and $\Delta_K$ represents the $K$-dimensional probability simplex. This soft assignment enables a probabilistic view of environment membership for each sample, facilitating robust environment-aware learning.

To support environment-aware learning with soft cluster assignments, we design a soft grouping mechanism for mini-batches Given a batch $\mathcal{B}$ of input samples $X = \{x, y_w, y_l\}$, and the corresponding soft cluster probabilities $p_{ik}$, which represents the soft membership of sample $i$ over $K$ latent environments, we aim to construct a soft representation for each environment. For a given cluster (environment) $k$, a soft aggregated representation $\bar{x}^{(k)}$ is computed as a weighted average of the inputs across the batch, where the cluster probabilities give the weights:

$$\bar{x}^{(k)} = \frac{\sum_{i=1}^{\mathcal{B}} p_{ik} \cdot x_i}{\sum_{i=1}^{\mathcal{B}} p_{ik}}, \quad \text{for } k \in \{1, \dots, K\}. \quad (9)$$

Here, $K$ denotes the *number of clusters discovered by DB-SCAN* on the current batch representations (excluding outliers labeled as noise), and thus is *data-driven rather than*

*manually specified.* In practice, DBSCAN determines $K$ implicitly through its hyperparameters, which control neighborhood density and robustness to noise. This formulation ensures that each environment-specific representation integrates information from all samples in the batch, proportionally weighted by their soft assignments to the corresponding environment. Such a soft grouping approach allows the model to learn disentangled environment-invariant features in a fully differentiable and data-efficient manner. Furthermore, we integrate the soft-clustering-based environment inference with the causal objective in Eq. (6).

Given the discovered $K$ clusters, each sample $z_i$ is assigned a soft membership distribution $p(E = k \mid z_i) = p_{i,k}$. We use these soft assignments to *estimate* the environment prior $p(E)$ from data. In practice, we approximate $p(E)$ by a mini-batch Monte Carlo estimator, i.e., the average membership probability across the batch: $\hat{p}(E = k) = \frac{1}{N} \sum_{i=1}^{N} p(E = k \mid Z_i)$. The approximate target distribution in Eq. (6) can be rewritten as: $p_\theta(Y \mid do(X)) \approx \sum_{k=1}^{K} p_\theta(Y \mid x, E = k) \cdot \hat{p}(E = k)$. Combining the back-door adjustment criterion, we weight the environment-conditioned preference model $p(Y \mid X, E = k)$ by the inferred environmental distribution $p(E)$, thereby approximating the causal objective $p(Y \mid do(x))$. Notably, the soft-clustering module is co-optimized with the main model parameters during training; this joint optimization allows the model to progressively refine the environment partition according to the empirical data distribution and task objectives, effectively mitigating the adverse effects of suboptimal initialization.

**Policy Invariant Learning**. We adopt soft clustering to assign pseudo-environment labels to samples, grouping together those affected by similar environmental factors. To learn stable preference patterns within groups and to regularize the global distribution across groups—thereby preventing DPO from overfitting to any particular environment when aligning the LLM to human preferences, we further introduce the principle of invariant learning, minimizing the discrepancies among these subsets during DPO optimization. Specifically, we use the Maximum Mean Discrepancy (MMD) metric to minimize the differences among outputs across environments, thereby yielding the overall optimization objective of CausalDPO:

$$\min_\theta \left\{ \mathcal{L}_{\text{DPO}}(\theta) + \lambda \cdot \text{MMD}(p_m, p_{m'}) \right\},$$

$$\mathcal{L}_{\text{DPO}}(\theta) = -\mathbb{E}_{(x,y_w,y_l)\sim\mathcal{O}} \left[ \log \sigma\big(r_\theta(x, y_w) - r_\theta(x, y_l)\big) \right],$$

$$\text{MMD}(p_m, p_{m'}) = \sum_{m,m'\in\mathcal{E}_{\text{train}}} \text{MMD}\Big(p_m\big(\pi_\theta(x, y_w) \mid do(x)\big),$$

$$p_{m'}\big(\pi_\theta(x, y_w) \mid do(x)\big)\Big). \tag{10}$$

Where $\mathcal{L}_{\text{DPO}}$ denotes the DPO loss, as formally defined in

Eq. (1), $\lambda$ is a hyperparameter that balances the preference learning objective and the invariance regularization term. The reward function is defined as $r_\theta(x, y) = \beta \log \frac{\pi_\theta(x,y)}{\pi_{\text{ref}}(x,y)}$, The indices $m$ and $m'$ range over the pseudo-environment set $\mathcal{E}_{\text{train}}$, derived from soft clustering over the training data. The term $p_m(z \mid do(x))$ denotes the distribution of an output representation $z$ (e.g., the policy score or hidden-state embedding induced by $\pi_\theta$ for $(x, y_w)$) under the causal intervention $do(x)$ within pseudo-environment $m$, characterizing how the model responds to the same intervened input across environments. The reason CausalDPO uses MMD as the regularization term is that it measures and minimizes discrepancies in the model's output distributions across environments, encouraging the model to ignore environment-specific noise or confounders and focus on features that are stable across all environments and truly reflect causal relationships (Modeste & Dombry, 2024). Current approaches (Wu et al., 2024c; Yang et al., 2022) based on invariant learning for OOD generalization commonly hinge on two fundamental assumptions: **invariance**, referring to the requirement that the learned representation yields consistent and optimal predictive performance across diverse environments; and **sufficiency**, which stipulates that the representation encodes all the necessary information for accurately predicting the target variable. We establish the optimality condition for Eq. (10) through the following proposition:

**Proposition 3.2** (Invariant and sufficient preference policy via CausalDPO.). *Let a preference policy model $\pi_\theta(y \mid x)$ be trained across environments $\mathcal{E}_{train} = \{\mathcal{E}_1, \ldots, \mathcal{E}_M\}$ using preference data $(x, y_w, y_l)$, where $y_w \succ y_l$ (denoting that $y_w$ is strictly preferred over $y_l$). We show that minimizing the objective in Eq. (10) induces a policy that satisfies invariance and sufficiency conditions:*

***Invariance:*** *The learned policy satisfies environment-invariant behavior over the optimal intervention distribution $do(x), \forall m, m' \in \mathcal{E}_{train}$, we have:*

$$p_m\left(\pi_\theta(x, y_w) \mid do(x)\right) = p_{m'}\left(\pi_\theta(x, y_w) \mid do(x)\right), \tag{11}$$

*as $\text{MMD} \to 0$, the policy distribution becomes invariant across environments.*

***Sufficiency.*** *The policy retains* sufficient *discriminative power for preferences: for samples $(x, y_w, y_l) \sim \mathcal{D}_{train}$, we have, with probability 1 under $\mathcal{D}_{train}$, $\pi_\theta(y_w \mid x) \gg \pi_\theta(y_l \mid x)$, where $\gg$ denotes strict dominance in log-probability. This follows from the DPO objective's explicit optimization of the preference log-ratio,*

$$\Delta r(x) = \log \frac{\pi_\theta(y_w \mid x)}{\pi_\theta(y_l \mid x)}, \tag{12}$$

*which preserves the model's discriminative capacity even under invariance constraints.*

**Proposition 3.3** (Generalization Bound for CausalDPO.).
*Let $\pi_\theta \in \Pi$ denote the learned policy parameterized by $\theta$, where $\Pi$ is the policy class. Let $p(e)$ and $q(e)$ denote the test and training environment distributions, respectively, over the environment space $\mathcal{E}$. Assume the function $f : \mathcal{E} \to \mathbb{R}$ belongs to a reproducing kernel Hilbert space (RKHS) $\mathcal{H}_k$ with kernel $k$, and satisfies $\|f\|_{\mathcal{H}_k} \leq C$ for some $C > 0$. Let $\ell : \mathbb{R} \times \mathcal{E} \to \mathbb{R}$ be a loss function that is $L_\ell$-Lipschitz in its first argument and bounded by $B > 0$. Then, with probability at least $1 - \delta$ over the draw of $n$ training environments, the generalization error for the risk $R_e(\pi_\theta) = \mathbb{E}_{a \sim \pi_\theta}[\ell(f(e, a), e)]$ satisfies:*

$$\sup_{e \in \mathcal{E}_{train}} \left| R_e(\pi_\theta) - \hat{R}_e(\pi_\theta) \right| \leq 2\mathcal{R}_n(\Pi) + B\sqrt{\frac{2\log(2/\delta)}{n}} \\ + L_\ell \cdot MMD_{\mathcal{H}_k}(p, q), \tag{13}$$

*where $\hat{R}_e(\pi_\theta)$ is the empirical risk over $n$ training environments $e_i \sim q(e)$. $\mathcal{R}_n(\Pi)$ is the empirical Rademacher complexity of the policy class $\Pi$ over $n$ samples. The MMD between two distributions $p$ and $q$ in the reproducing kernel Hilbert space $\mathcal{H}_k$ is defined as*

$$MMD_{\mathcal{H}_k}(p, q) = \sup_{\|f\|_{\mathcal{H}_k} \leq 1} \left| \mathbb{E}_{e \sim p}[f(e)] - \mathbb{E}_{e \sim q}[f(e)] \right|. \tag{14}$$

Proposition A.2 demonstrates the effectiveness of the optimization objective Eq. (10) in improving the generalization performance of DPO on OOD data, thereby enhancing the theoretical soundness and reliability of our method. Proposition A.3 further shows that the generalization error bound of the model under distribution shifts is controllable. This conclusion can also be interpreted from the perspective of SCM: optimizing Eq. (10) not only helps eliminate spurious dependencies caused by environmental confounders but also strengthens the ability of DPO to model invariant causal mechanisms across multiple environments. Detailed proofs of the above propositions are provided in Appendix A.3 and A.4. The process pseudocode of fine-tuning CausalDPO is shown in Algorithm 1.

### 3.4. Time Complexity Analysis.

Table 6 reports the algorithmic complexity and per-epoch time for DPO, its variants, and CausalDPO on the Book-Crossing dataset. CausalDPO exhibits a complexity of $\mathcal{O}(BL^2 d + B^2)$, where the quadratic term in $B$ arises from MMD-based pairwise environment-consistency computations. Empirically, CausalDPO requires 2971 s per epoch versus 2482 s for DPO ($+19.70\%$), yet achieves an average performance improvement of $205.9\%$ over DPO, indicating a favorable compute accuracy trade-off. The additional overhead primarily stems from (i) soft clustering for latent-environment inference and (ii) regularization that enforces

distributional consistency across environments. Among the baselines, SimPO is the fastest owing to its omission of reference model alignment.

### 3.5. Discussion

This subsection investigates two key questions: (i) **the correspondence between clustered environments and the true (unobserved) confounders; and (ii) the behavior of CausalDPO when clustering fails or the OOD assumption does not hold**. To encourage alignment between the inferred environments and the true confounders, CausalDPO incorporates two design choices: (1) it estimates a pseudo-environment distribution $p(\hat{E} \mid z)$ using soft clustering followed by a softmax mapping, and updates the clusters dynamically during training to avoid biases from fixed assignments; and (2) it adopts a soft-assignment scheme under which each example's environment membership converges probabilistically, thereby better reflecting latent shifts in the data. Even when the pseudo environments $\hat{E}$ are not perfectly aligned with the true environments $E$, the MMD regularizer reduces distributional discrepancies across pseudo environments, smoothing the learned policy outputs and attenuating environment-specific noise, which in turn improves robustness. Table 3 reports a comparison among DPO variants under IID conditions; CausalDPO remains superior even when OOD assumptions are not satisfied.

### 3.6. Experimental settings

**Datasets**. We follow the experimental setup of prior works (Wang et al., 2024a; Zhao et al., 2025a) and conduct systematic evaluations under four distribution shift scenarios: popularity shift, temporal shift, exposure shift, and mixed shift. To assess the robustness of CausalDPO, we test it on three standard benchmarks: Yelp2018 (Chitla & Kalluri, 2024), Movielens-10M (Zhao et al., 2024), and Book-Crossing (Samridhi et al., 2024). Detailed dataset specifications and preprocessing steps are provided in Appendix B.1. **Baselines**. Following the existing research (Gao et al., 2024), we systematically compare three major categories of methods in the recommendation field: (1) traditional recommendation models: SASRec (Kang & McAuley, 2018); (2) SFT-based methods: BIGRec (Bao et al., 2025), RW (Jiang et al., 2024), and $D^3$ (Bao et al., 2024); and (3) DPO-optimized models: DMPO (Bai et al., 2024), SDPO (Chen et al., 2024), RosePO (Liao et al., 2024), and SPRec (Gao et al., 2024). We show the details of each model in Appendix B.2. **Evaluation**. We directly follow existing works (Bao et al., 2024; Chen et al., 2024; Bao et al., 2025; Gao et al., 2024; Liao et al., 2024) by using prompts to guide LLMs in generating predicted items based on the user's historical interaction sequence. We then perform scoring and ranking across the entire item space and map the predicted items to specific corresponding items in the dataset. The main evaluation metrics employed

were HR@K and NDCG@K, where K is set to 10 and 20. **Implementation**. The detailed implementation specifics are provided in Appendix B.3.

### 3.7. Comparative Results

**Evaluation on popularity shift**. As shown in Table 1, CausalDPO significantly outperforms all baselines on the Yelp2018 dataset, with an average relative gain of 22.29% over the strongest competitor. This substantial improvement underscores its effectiveness in handling popularity-based distribution shifts, especially in modeling preferences for long-tail items. **Evaluation on temporal shift**. Table 1 compares the test performance of various models on the Movielens-10M dataset. Our proposed CausalDPO significantly outperforms all baselines across key metrics, with an average performance gain of 24.06%, demonstrating its strength in modeling temporal dependencies. By leveraging causal structure modeling and invariant causal learning, CausalDPO captures stable causal relations, mitigates temporal shifts, and improves generalization. **Evaluation on exposure shift**. In real-world scenarios, users are typically exposed to a limited set of items, leading to non-random missing data, known as exposure shift. Experimental results in Table 1 on a synthesized fully exposed version of the Book-Crossing dataset show that CausalDPO consistently outperforms baselines, with performance gains of 8.47% to 23.33%, validating its effectiveness in mitigating the impact of exposure shift on generative recommendation models. **Evaluation on mixed shift**. Table 4 reports the experimental results on hybrid biased data. The results show that CausalDPO consistently outperforms all baseline models under both popularity and noisy shifts, demonstrating its strong ability to ensure OOD generalization for generative recommendation models under multiple distributional shifts. Moreover, Table 3 shows that CausalDPO remains competitive under IID while retaining strong OOD generalization; meanwhile, SASRec excels on sparse data (e.g., Book-Crossing) because self-attention, with positional encoding and localized attention learns effective user representations from sparse interaction sequences.

### 3.8. Ablation Studies

We present the results of ablation studies in Table 8. Specifically, **w/o SFT** denotes skipping the supervised fine-tuning stage and directly performing preference alignment; **w/o CausalDPO** denotes removing preference alignment and applying only supervised fine-tuning; and **w/o Both** indicates disabling both stages. Based on the results, we make the following key observations: (1) On both datasets, the **w/o SFT** variant yields the worst performance, even inferior to the **w/o Both** setting, highlighting the critical role of supervised fine-tuning in establishing a strong base for LLM-based recommendation. (2) Removing CausalDPO

while retaining supervised fine-tuning leads to a significant performance drop across both datasets, demonstrating the importance of CausalDPO in mitigating spurious correlations caused by environmental confounders and improving generalization under distribution shift. Overall, the ablation studies provide strong evidence for the effectiveness and necessity of each proposed module in CausalDPO. In addition, Table 5 compares several DPO variants originally designed for non-recommendation tasks (e.g., SimPO (Meng et al., 2024), CPO (Guo et al., 2024)). These methods exhibit poor OOD generalization compared to CausalDPO. We further extend these variants by integrating the CausalDPO framework as their backbone. The results show consistent improvements in OOD performance, suggesting the general applicability of CausalDPO. A detailed analysis of performance and time complexity is shown in Appendix B.4. Moreover, in Appendix B.4, we explore the performance of CausalDPO using LLMs of varying parameter scales as the backbone.

### 3.9. Qualitative Interpretation of Pseudo-Environment Clustering.

To understand what the inferred pseudo-environments $\hat{E}$ capture, we conduct qualitative analyses under two representative shift factors: **popularity bias** on Yelp2018 and **temporal drift** on ML-10M. Figure 2 (left) reports NDCG@10 from head to tail groups (G1→G5). While all methods degrade as popularity decreases, CausalDPO exhibits a flatter drop and consistently outperforms the baselines, with especially clear gains on long-tail groups (G4–G5). Figure 2 (middle-left) shows performance over time (T1→T5), where CausalDPO remains notably more stable and the baselines deteriorate more sharply in later periods, indicating stronger robustness to preference drift. We further visualize low-dimensional projections of learned representations colored by pseudo-environment assignments (Figure 2, right). On Yelp2018, clusters correlate with popularity strata and show a head-to-tail transition pattern, reflecting popularity-driven exposure heterogeneity. On ML-10M, samples form stage-wise groupings with observable shifts, characterizing temporal preference drift. Overall, these results suggest that CausalDPO identifies meaningful latent environment structure in the representation space, supporting cross-environment invariance regularization and improving OOD generalization under distribution shifts.

## 4. Related Works

**LLM for Recommendation**. The application of LLMs in recommender systems (Kim et al., 2024; Fan, 2024; Li et al., 2023b; Lin et al., 2025; Wu et al., 2024b; Liu et al., 2025b;a; 2024b) can be broadly categorized into two approaches: augmentation-based methods and generation-based meth-

*Table 1.* The performance (%) comparison between the baselines and CausalDPO on the three datasets with three **OOD distributions**. We highlight the methods with the **best** and second-best results.

| Methods | Yelp2018 | | | | ML-10M | | | | Book-Crossing | | | |
|---|---|---|---|---|---|---|---|---|---|---|---|---|
| | H@10 | N@10 | H@20 | N@20 | H@10 | N@10 | H@20 | N@20 | H@10 | N@10 | H@20 | N@20 |
| SASRec | 1.01 | 0.43 | 1.51 | 0.61 | 0.50 | 0.02 | 0.84 | 0.30 | 0.33 | 0.21 | 0.68 | 0.21 |
| BIGRec | 0.77 | 0.50 | 1.28 | 0.62 | 0.53 | 0.29 | 0.76 | 0.35 | 0.53 | 0.30 | 0.98 | 0.43 |
| RW | 0.51 | 0.42 | 1.02 | 0.56 | 0.50 | 0.28 | 0.74 | 0.30 | 0.59 | 0.29 | 0.91 | 0.41 |
| D3 | 0.63 | 0.38 | 1.15 | 0.51 | 0.50 | 0.25 | 0.80 | 0.34 | 0.50 | 0.27 | 0.94 | 0.40 |
| DMPO | 0.76 | 0.26 | 1.54 | 0.43 | 0.44 | 0.20 | 0.81 | 0.29 | 0.19 | 0.10 | 0.39 | 0.15 |
| SDPO | 0.76 | 0.54 | 1.28 | 0.66 | 0.55 | 0.30 | 0.87 | 0.38 | 0.49 | 0.25 | 0.74 | 0.31 |
| RosePO | 0.82 | 0.47 | 1.36 | 0.60 | 0.51 | 0.26 | 0.83 | 0.33 | 0.40 | 0.20 | 0.71 | 0.36 |
| SPRec | 1.02 | 0.61 | 1.64 | 0.79 | 0.55 | 0.30 | 0.71 | 0.36 | 0.39 | 0.18 | 0.69 | 0.25 |
| **Ours** | **1.53** | **0.77** | **1.79** | **0.82** | **0.61** | **0.35** | **1.02** | **0.45** | **0.64** | **0.37** | **1.08** | **0.48** |

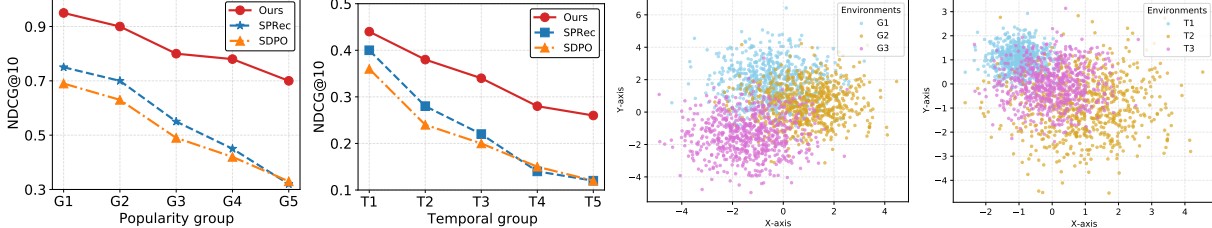

*Figure 2.* Further study on performance under distribution shifts and clustering Visualization.

ods. Augmentation-based methods (Zhao et al., 2025b; Liu et al., 2024a; Ren et al., 2024; Wei et al., 2024; Dang et al., 2026b;a) leverage the extensive world knowledge encoded in LLMs to generate additional semantic information, such as user profiles or completed interaction sequences, through carefully designed prompts. This information is used to enhance traditional ID-based models by enriching feature representations and improving the modeling of user preferences. In addition, some studies (Cui et al., 2024; Du et al., 2025; Li et al., 2023a) have explored using knowledge distillation to transfer the reasoning capabilities of LLMs to traditional recommendation models, aiming to improve their performance and generalization ability. In contrast, generation-based methods (Zhang et al., 2025; Hua et al., 2023; Geng et al., 2022; Wang et al., 2024b) incorporate collaborative signals into the input context and employ techniques such as supervised fine-tuning or preference alignment to predict the next item interaction, thereby moving beyond the need for explicit ID-based modeling in traditional recommendation paradigms.

**Preference Alignment in LLM-based Recommendation**. Reinforcement learning from human feedback (RLHF) (Christiano et al., 2017) has emerged as the dominant paradigm for aligning large language models (LLMs) with human values. However, due to the challenges associated with designing complex reward functions in RLHF, researchers have proposed Direct Preference Optimization (DPO) (Rafailov et al., 2023), which derives the optimal policy in closed form and directly optimizes LLMs using

human preference data. Building upon this research foundation, researchers have developed multiple DPO variant methodologies (Wang et al., 2024c; Richemond et al., 2024). Since DPO explicitly models user preferences in a manner that aligns with the fundamental objective of recommendation, which is to rank preferred items higher, it has been naturally adopted in recommender systems to enhance the performance of approaches based on LLMs (Bai et al., 2024; Chen et al., 2024; Liao et al., 2024; Gao et al., 2024; Deng et al., 2025). This study further reveals the potential adverse effects of data distribution shifts on the preference alignment process in DPO and proposes a causal intervention approach to enhance the OOD generalization ability of LLM-based recommender systems.

## 5. Conclusion

In this paper, we first reveal that during preference alignment using Direct Preference Optimization, LLM-based recommendation systems tend to amplify spurious correlations caused by environmental confounders, significantly undermining their OOD generalization. To address this issue, we propose CausalDPO, a causality-aware extension to DPO based on the principle of invariant causal learning. CausalDPO eliminates the influence of environmental confounders and enforces distributional consistency across different environments, thereby enhancing the model's generalization to unseen distributions. Extensive experiments show that our method consistently improves generative rec-

ommendation under diverse distribution shifts.

## Acknowledgements

This work is partially supported by the National Natural Science Foundation of China under Grant No. 62576083. We extend special thanks to the National Supercomputer Center in Guangzhou for their computational support.

## Impact Statement

This paper presents work whose goal is to advance the field of machine learning. There are many potential societal consequences of our work, none of which we feel must be specifically highlighted here.

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

# A. Theoretical Proof

## A.1. Proof of Proposition 1: DPO Amplifies Spurious Correlations and Hinders Generalization

We demonstrate that optimizing the Direct Preference Optimization (DPO) objective under biased training data increases the model's reliance on spurious environmental variables $E$, thereby impairing generalization under distribution shifts. The proof proceeds by defining the DPO loss, modeling the effect of spurious correlations, analyzing the gradient dynamics, showing overfitting to spurious features, and quantifying the generalization error.

Consider the DPO scoring function defined as $r_\theta(x, y) = \beta \log \frac{\pi_\theta(y|x)}{\pi_{\text{ref}}(y|x)}$, where $\pi_\theta(y|x)$ is the model's conditional probability, $\pi_{\text{ref}}(y|x)$ is a reference policy, and $\beta > 0$ is a scaling factor. The DPO loss is given by:

$$\mathcal{L}_{\text{DPO}} = -\mathbb{E}_{(x,y_w,y_l)\sim D} \left[ \log \sigma \left( r_\theta(x, y_w) - r_\theta(x, y_l) \right) \right], \tag{15}$$

where $D$ is the dataset of preference tuples $(x, y_w, y_l)$, with $y_w$ preferred over $y_l$ for input $x$, and $\sigma(z) = (1 + e^{-z})^{-1}$ is the sigmoid function. This loss encourages $\pi_\theta(y_w|x) > \pi_\theta(y_l|x)$ by maximizing the difference $r_\theta(x, y_w) - r_\theta(x, y_l)$.

Assume the model's log-probability decomposes as:

$$\log \pi_\theta(y|x) = w_E \cdot f_E(y, E) + w_{\text{rest}} \cdot f_{\text{rest}}(y, X_{\text{rest}}) + b, \tag{16}$$

where $x = (E, X_{\text{rest}})$, $E$ represents environmental features, and $X_{\text{rest}}$ denotes task-relevant features. The term $f_E(y, E)$ captures alignment with spurious environmental variables, while $f_{\text{rest}}(y, X_{\text{rest}})$ encodes task-relevant information, with weights $w_E$, $w_{\text{rest}}$, and bias $b$. In the biased training distribution, suppose:

$$p_{\text{train}}(E \mid y_w) > p_{\text{train}}(E \mid y_l), \tag{17}$$

implying that $E$ is spuriously correlated with the preferred output $y_w$. This leads to:

$$\mathbb{E}_D[f_E(y_w, E) - f_E(y_l, E)] > 0, \tag{18}$$

indicating that the spurious feature $f_E$ is more strongly associated with $y_w$ than $y_l$ in the training data.

To understand the effect of DPO optimization, we compute the gradient of the loss with respect to $w_E$:

$$\nabla_{w_E}\mathcal{L}_{\text{DPO}} = -\mathbb{E}_{(x,y_w,y_l)\sim D} \left[ (1 - \sigma(r_\theta(x, y_w) - r_\theta(x, y_l))) \cdot \beta(f_E(y_w, E) - f_E(y_l, E)) \right]. \tag{19}$$

Since $1 - \sigma(z) > 0$ for all $z$ and $\mathbb{E}_D[f_E(y_w, E) - f_E(y_l, E)] > 0$, the gradient satisfies:

$$\nabla_{w_E}\mathcal{L}_{\text{DPO}} < 0. \tag{20}$$

With gradient descent updates $w_E^{(t+1)} = w_E^{(t)} - \eta \nabla_{w_E}\mathcal{L}_{\text{DPO}}$ for learning rate $\eta > 0$, it follows that:

$$w_E^{(t+1)} > w_E^{(t)}, \tag{21}$$

showing that the weight $w_E$ increases over training iterations, amplifying reliance on the spurious feature $E$.

The log-odds preference score is:

$$\Delta_E = \log \frac{\pi_\theta(y_w|x)}{\pi_\theta(y_l|x)} = r_\theta(x, y_w) - r_\theta(x, y_l) \approx \beta w_E(f_E(y_w, E) - f_E(y_l, E)). \tag{22}$$

After $T$ iterations, assuming a constant expected gradient for simplicity, we approximate:

$$w_E^{(T)} \approx w_E^{(0)} + \eta T \cdot \beta \cdot \mathbb{E}_D[f_E(y_w, E) - f_E(y_l, E)]. \tag{23}$$

Since $\mathbb{E}_D[f_E(y_w, E) - f_E(y_l, E)] > 0$, the weight $w_E^{(T)}$ grows linearly with $T$, leading to:

$$\Delta_E \propto T \cdot \mathbb{E}_D[f_E(y_w, E) - f_E(y_l, E)]. \tag{24}$$

Thus, DPO progressively overfits to the spurious feature $E$, as the preference score becomes increasingly dominated by $f_E$.

To assess generalization, consider a test distribution $P_{\text{test}}$ where the spurious correlation between $E$ and $y$ differs from $P_{\text{train}}$. The generalization error is approximated as:

$$\text{GenErr}(\theta) = \mathbb{E}_{P_{\text{test}}(x)} \left| \mathbb{E}_{P_{\text{test}}(y|x)} \mathbb{E}_{P_{\text{test}}(E|x)} [w_E f_E(y, E)] - \mathbb{E}_{P_{\text{train}}(y|x,E)} [w_E f_E(y, E)] \right|. \tag{25}$$

Assuming $P_{\text{test}}(y|x) \approx P_{\text{train}}(y|x)$, the error simplifies to:

$$\text{GenErr}(\theta) \approx |w_E| \cdot \mathbb{E}_{P_{\text{test}}(x)} \left| \mathbb{E}_{P_{\text{test}}(E|x)} [f_E(y, E)] - \mathbb{E}_{P_{\text{train}}(E|x)} [f_E(y, E)] \right|. \tag{26}$$

Suppose $|f_E(y, E)| \leq C$ for some constant $C$. Using the total variation distance, $\text{TV}(P_1, P_2) = \frac{1}{2} \int |P_1(E) - P_2(E)| dE$, we bound:

$$\left| \mathbb{E}_{p_{\text{test}}(E|x)} [f_E] - \mathbb{E}_{p_{\text{train}}(E|x)} [f_E] \right| \leq \int |f_E(y, E)| \cdot |p_{\text{test}}(E|x) - p_{\text{train}}(E|x)| dE \tag{27}$$

$$\leq 2C \cdot \text{TV}(p_{\text{train}}(E|x), p_{\text{test}}(E|x)). \tag{28}$$

Thus, the generalization error is bounded by:

$$\text{GenErr}(\theta) \leq 2C|w_E| \cdot \mathbb{E}_{p_{\text{test}}(x)} \left[ \text{TV}(p_{\text{train}}(E|x), p_{\text{test}}(E|x)) \right]. \tag{29}$$

When the model increasingly relies on environmental signals—i.e., the effective dependence on $E$ becomes stronger (e.g., $|w_E|$ increases)—and the conditional environment distribution shifts across domains, $P_{\text{train}}(E \mid x) \neq P_{\text{test}}(E \mid x)$, the bound becomes looser and the generalization error can increase accordingly. In other words, if DPO training encourages the policy to exploit environment-induced spurious cues, this reliance may be amplified during optimization, leading to larger errors under distribution shifts and thus degraded generalization.

### A.2. Derivation for Equation 6

We follow the basic do-calculus rules in (Wu et al., 2024c). Consider a causal DAG $G$ over $\{E, X, Y\}$ (Fig. 1(c), right). Let $G_{\overline{X}}$ denote the mutilated graph obtained by deleting all incoming edges into $X$, and let $G_{\underline{X}}$ denote the mutilated graph obtained by deleting all outgoing edges from $X$. For any interventional distribution compatible with $G$, the do-calculus yields:

**(1) Action/Observation Exchange:**

$$P(Y \mid do(E), do(X)) = P(Y \mid do(E), X), \quad \text{if } (E \perp\!\!\!\perp Y \mid X)_{G_{\overline{X}\underline{E}}}. \tag{30}$$

**(2) Insertion/Deletion of Actions:**

$$P(Y \mid do(E), do(X)) = P(Y \mid do(E)), \quad \text{if } (X \perp\!\!\!\perp Y \mid E)_{G_{\overline{E}}}. \tag{31}$$

Based on the above, Eq. (6) follows by standard backdoor adjustment:

$$
\begin{aligned}
p_\theta(Y \mid do(X)) &= \sum_e p_\theta(Y \mid do(X), E = e) \, p_\theta(E = e) \\
&= \sum_e p_\theta(Y \mid X, E = e) \, p_\theta(E = e) \quad \text{(by Rule 1, since } (E \perp\!\!\!\perp Y \mid X)_{G_{\overline{X}}}) \\
&= \sum_e p_\theta(Y \mid X, E = e) \, p_\theta(E = e \mid do(X)) \\
&= \sum_e p_\theta(Y \mid X, E = e) \, p_\theta(E = e) \quad \text{(since } E \text{ is not affected by } do(X), \text{ i.e., } p(E \mid do(X)) = p(E)) \\
&= \mathbb{E}_{E \sim p_\theta(E)} [p_\theta(Y \mid X, E)].
\end{aligned}
\tag{32}
$$

### A.2.1. PROBLEM SETUP

We interpret $\pi_\theta(x, y) = p_\theta(y \mid x)$ as a stochastic policy in a policy gradient framework. The objective is:

$$J(\theta) = \mathcal{L}_{\text{DPO}} + \lambda \cdot \text{MMD}(p_m, p_{m'}), \tag{33}$$

with policy gradient:

$$\nabla_\theta J(\theta) = \nabla_\theta \mathcal{L}_{\text{DPO}} + \lambda \nabla_\theta \text{MMD}(p_m, p_{m'}). \tag{34}$$

The Invariance Property requires:

$$p_m(\pi_\theta(x, y_w) \mid do(x)) = p_{m'}(\pi_\theta(x, y_w) \mid do(x)), \quad \forall m, m' \in \mathcal{E}_{\text{train}}, \tag{35}$$

ensuring the policy ignores spurious environmental variations. The Sufficient Condition requires that $\pi_\theta$ captures true causal relationships, maintaining low $\mathcal{L}_{\text{DPO}}$.

## A.3. Proof of Invariant and Sufficient Preference Policy via CausalDPO

**Proposition A.2** (Invariant and sufficient preference policy via CausalDPO.). *Let a preference policy model $\pi_\theta(y \mid x)$ be trained across environments $\mathcal{E}_{train} = \{\mathcal{E}_1, \ldots, \mathcal{E}_M\}$ using preference data $(x, y_w, y_l)$, where $y_w \succ y_l$ (denoting that $y_w$ is strictly preferred over $y_l$). We show that minimizing the objective in Eq. (10) induces a policy that satisfies invariance and sufficiency conditions:*

*[leftmargin=\*]*

- ***Invariance:*** *The learned policy satisfies environment-invariant behavior over the optimal intervention distribution $do(x)$:*

$$\forall m, m' \in \mathcal{E}_{train}, \quad p_m\left(\pi_\theta(x, y_w) \mid do(x)\right) = p_{m'}\left(\pi_\theta(x, y_w) \mid do(x)\right),$$

*as $\text{MMD} \to 0$, the policy distribution becomes invariant across environments.*

- ***Sufficiency:*** *The policy maintains* sufficiency *for preference discrimination, satisfying:*

$$\pi_\theta(y_w \mid x) \gg \pi_\theta(y_l \mid x) \quad \text{almost surely for } (x, y_w, y_l) \sim \mathcal{D}_{train}, \tag{36}$$

*where $\gg$ denotes strict dominance in log-probability. This follows from the DPO objective's explicit optimization of the preference log-ratio:*

$$\Delta r(x) = \log\left(\frac{\pi_\theta(y_w \mid x)}{\pi_\theta(y_l \mid x)}\right), \tag{37}$$

*which preserves the model's discriminative capacity even under invariance constraints.*

### A.3.1. POLICY GRADIENT DERIVATION

**DPO Gradient.** The DPO loss is:

$$\mathcal{L}_{\text{DPO}} = -\mathbb{E}_{(x, y_w, y_l) \sim \mathcal{O}} \left[\log \sigma\left(\Delta r\right)\right], \tag{38}$$

where:

$$\Delta r = r_\theta(x, y_w) - r_\theta(x, y_l) = \beta \log \frac{\pi_\theta(x, y_w)}{\pi_\theta(x, y_l)} + \beta \log \frac{\pi_{\text{ref}}(x, y_l)}{\pi_{\text{ref}}(x, y_w)}. \tag{39}$$

The gradient is:

$$\nabla_\theta \mathcal{L}_{\text{DPO}} = -\mathbb{E}_{(x, y_w, y_l)} \left[(1 - \sigma(\Delta r))\nabla_\theta \Delta r\right], \tag{40}$$

since $\nabla_\theta \log \sigma(z) = (1 - \sigma(z))\nabla_\theta z$. Compute:

$$\nabla_\theta \Delta r = \beta \left(\nabla_\theta \log \pi_\theta(x, y_w) - \nabla_\theta \log \pi_\theta(x, y_l)\right) = \beta \left(\frac{\nabla_\theta \pi_\theta(x, y_w)}{\pi_\theta(x, y_w)} - \frac{\nabla_\theta \pi_\theta(x, y_l)}{\pi_\theta(x, y_l)}\right). \tag{41}$$

Thus:

$$\nabla_\theta \mathcal{L}_{\text{DPO}} = -\beta \mathbb{E}_{(x, y_w, y_l)} \left[(1 - \sigma(\Delta r))\left(\frac{\nabla_\theta \pi_\theta(x, y_w)}{\pi_\theta(x, y_w)} - \frac{\nabla_\theta \pi_\theta(x, y_l)}{\pi_\theta(x, y_l)}\right)\right]. \tag{42}$$

This gradient increases $\pi_\theta(x, y_w)$ and decreases $\pi_\theta(x, y_l)$, aligning the policy with preferences.

**MMD Gradient.** The MMD term is:

$$\mathrm{MMD}(p_m, p_{m'}) = \sum_{m,m' \in \mathcal{E}_{\text{train}}} \mathrm{MMD}(p_m, p_{m'}), \tag{43}$$

where:

$$\mathrm{MMD}^2(p_m, p_{m'}) = \mathbb{E}_{p_m, p_m}[k(z, z')] + \mathbb{E}_{p_{m'}, p_{m'}}[k(z, z')] - 2\mathbb{E}_{p_m, p_{m'}}[k(z, z')], \tag{44}$$

with $z = \pi_\theta(x, y_w) \mid do(x)$, and $k$ is a characteristic kernel (e.g., Gaussian). The empirical gradient is:

$$
\begin{aligned}
\nabla_\theta \mathrm{MMD}^2 \approx \sum_{m,m'} &\left[ \frac{1}{N_m^2} \sum_{i=1}^{N_m} \sum_{j=1}^{N_m} \frac{\partial k(z_i^m, z_j^m)}{\partial z_i^m} \nabla_\theta z_i^m + \frac{1}{N_{m'}^2} \sum_{i=1}^{N_{m'}} \sum_{j=1}^{N_{m'}} \frac{\partial k(z_i^{m'}, z_j^{m'})}{\partial z_i^{m'}} \nabla_\theta z_i^{m'} \right. \\
&\left. - \frac{2}{N_m N_{m'}} \sum_{i=1}^{N_m} \sum_{j=1}^{N_{m'}} \left( \frac{\partial k(z_i^m, z_j^{m'})}{\partial z_i^m} \nabla_\theta z_i^m + \frac{\partial k(z_i^m, z_j^{m'})}{\partial z_j^{m'}} \nabla_\theta z_j^{m'} \right) \right].
\end{aligned}
\tag{45}
$$

where $z_i^m = \pi_\theta(x, y_w)$. For a Gaussian kernel, $\frac{\partial k(z, z')}{\partial z} = -\frac{1}{\sigma^2} k(z, z')(z - z')$, aligning distributions $p_m$ and $p_{m'}$.

### A.3.2. INVARIANCE PROPERTY

The Invariance Property requires $p_m(\pi_\theta(x, y_w) \mid do(x)) = p_{m'}(\pi_\theta(x, y_w) \mid do(x))$. The MMD term minimizes:

$$\sum_{m,m'} \left\| \mathbb{E}_{p_m}[\phi(\pi_\theta)] - \mathbb{E}_{p_{m'}}[\phi(\pi_\theta)] \right\|_{\mathcal{H}}^2. \tag{46}$$

Since $k$ is characteristic, $\mathrm{MMD}(p_m, p_{m'}) = 0$ if and only if $p_m = p_{m'}$. The gradient $\nabla_\theta \mathrm{MMD}$ adjusts $\pi_\theta$ to reduce distributional discrepancies. For gradient descent:

$$\theta_{t+1} = \theta_t - \eta \left( \nabla_\theta \mathcal{L}_{\mathrm{DPO}} + \lambda \nabla_\theta \mathrm{MMD} \right), \tag{47}$$

a sufficiently large $\lambda$ ensures $\nabla_\theta \mathrm{MMD} \to 0$, implying:

$$\mathbb{E}_{p_m}[\phi(\pi_\theta)] = \mathbb{E}_{p_{m'}}[\phi(\pi_\theta)] \implies p_m = p_{m'}. \tag{48}$$

The intervention $do(x)$ isolates causal effects, ensuring $\pi_\theta$ depends only on $x$, satisfying the Invariance Property.

### A.3.3. SUFFICIENT CONDITION

The Sufficient Condition requires low $\mathcal{L}_{\mathrm{DPO}}$, ensuring preference alignment. The DPO gradient maximizes:

$$\log \frac{\pi_\theta(x, y_w)}{\pi_\theta(x, y_l)}, \tag{49}$$

derived from the Bradley-Terry model. For an expressive $\pi_\theta$, minimizing $\mathcal{L}_{\mathrm{DPO}}$ achieves:

$$\sigma(\Delta r) \to 1 \implies r_\theta(x, y_w) - r_\theta(x, y_l) \to \infty. \tag{50}$$

The total gradient balances:

$$\nabla_\theta \mathcal{L}_{\mathrm{DPO}} + \lambda \nabla_\theta \mathrm{MMD} = 0. \tag{51}$$

A tuned $\lambda$ ensures $\mathrm{MMD} \approx 0$ while $\mathcal{L}_{\mathrm{DPO}}$ remains low, as the MMD constraint preserves invariant features predictive of preferences. Convergence to a stationary point satisfies both conditions, assuming bounded gradients and sufficient data.

By jointly optimizing preference fidelity and environment invariance, the proposed objective ensures that the learned policy both (i) generalizes across training environments and (ii) captures causal user-item preference signals. Gradual dynamics naturally balance these two objectives during training, ensuring convergence with a robust and discriminative policy.

## A.4. Proof of Generalization Bound for CausalDPO

**Proposition A.3** (Generalization Bound for CausalDPO.). *Let $\pi_\theta \in \Pi$ denote the learned policy parameterized by $\theta$, where $\Pi$ is the policy class. Let $p(e)$ and $q(e)$ denote the test and training environment distributions, respectively, over the environment space $\mathcal{E}$. Assume the function $f : \mathcal{E} \to \mathbb{R}$ belongs to a reproducing kernel Hilbert space (RKHS) $\mathcal{H}_k$ with kernel $k$, and satisfies $\|f\|_{\mathcal{H}_k} \le C$ for some $C > 0$. Let $\ell : \mathbb{R} \times \mathcal{E} \to \mathbb{R}$ be a loss function that is $L_\ell$-Lipschitz in its first argument and bounded by $B > 0$. Then, with probability at least $1 - \delta$ over the draw of $n$ training environments, the generalization error for the risk $R_e(\pi_\theta) = \mathbb{E}_{a \sim \pi_\theta}[\ell(f(e, a), e)]$ satisfies:*

$$\sup_{e \in \mathcal{E}_{train}} \left| R_e(\pi_\theta) - \hat{R}_e(\pi_\theta) \right| \le 2\mathcal{R}_n(\Pi) + B\sqrt{\frac{2\log(2/\delta)}{n}} + L_\ell \cdot MMD_{\mathcal{H}_k}(p, q), \tag{52}$$

*where $\hat{R}_e(\pi_\theta)$ is the empirical risk over $n$ training environments $e_i \sim q(e)$. $\mathcal{R}_n(\Pi)$ is the empirical Rademacher complexity of the policy class $\Pi$ over $n$ samples. $MMD_{\mathcal{H}_k}(p, q) = \sup_{\|f\|_{\mathcal{H}_k} \le 1} |\mathbb{E}_{e \sim p}[f(e)] - \mathbb{E}_{e \sim q}[f(e)]|$ is the maximum mean discrepancy between $p$ and $q$ in $\mathcal{H}_k$.*

*Proof.* We decompose the generalization error as:

$$\left| R_e(\pi) - \hat{R}_e(\pi) \right| = \left| \mathbb{E}_{e \sim p}[g_\pi(e)] - \frac{1}{n}\sum_{i=1}^{n} g_\pi(e_i) \right|,$$

where $g_\pi(e) := \mathbb{E}_{a \sim \pi}[\ell(f(e, a), e)]$. This can be split into:

$$|\mathbb{E}_p[g_\pi(e)] - \mathbb{E}_q[g_\pi(e)]| + \left| \mathbb{E}_q[g_\pi(e)] - \frac{1}{n}\sum_{i=1}^{n} g_\pi(e_i) \right|.$$

**(I) Bounding the Empirical Estimation Term.** Let $\mathcal{F} = \{g_\pi : \pi \in \Pi\}$. Since $|\ell| \le B$, we have $|g_\pi(e)| \le B$. Using Rademacher complexity bounds for real-valued function classes, with probability at least $1 - \delta/2$:

$$\sup_{\pi \in \Pi} \left| \mathbb{E}_q[g_\pi(e)] - \frac{1}{n}\sum_{i=1}^{n} g_\pi(e_i) \right| \le 2\mathcal{R}_n(\Pi) + B\sqrt{\frac{2\log(2/\delta)}{n}}.$$

**(II) Bounding the Distribution Shift Term.** For each fixed $a$, $f(e, a) \in \mathcal{H}_k$, and by the reproducing property:

$$f(e, a) = \langle f_a, k(e, \cdot) \rangle_{\mathcal{H}_k}, \quad \text{with } \|f_a\|_{\mathcal{H}_k} \le C.$$

By the $L_\ell$-Lipschitz property of $\ell$:

$$|g_\pi(e) - g_\pi(e')| \le L_\ell \cdot \mathbb{E}_{a \sim \pi}|f(e, a) - f(e', a)|.$$

Using the RKHS norm:

$$|f(e, a) - f(e', a)| \le \|f\|_{\mathcal{H}_k} \cdot \|k(e, \cdot) - k(e', \cdot)\|_{\mathcal{H}_k} \le 2C \cdot \sqrt{k(e, e) + k(e', e') - 2k(e, e')}.$$

Thus, $g_\pi$ inherits smoothness in $\mathcal{H}_k$, and:

$$|\mathbb{E}_p[g_\pi(e)] - \mathbb{E}_q[g_\pi(e)]| \le L_\ell \cdot \mathrm{MMD}_{\mathcal{H}_k}(p, q).$$

**(III) Final Bound.** Taking a union bound over the two terms, we conclude that with probability at least $1 - \delta$:

$$\left| R_e(\pi) - \hat{R}_e(\pi) \right| \le 2\mathcal{R}_n(\Pi) + B\sqrt{\frac{2\log(2/\delta)}{n}} + L_\ell \cdot \mathrm{MMD}_{\mathcal{H}_k}(p, q).$$

$\square$

## B. Experimental Settings and Supplementary Experiments

*Table 2.* Statistics of datasets.

| Dataset | Movielens-10M | Yelp2018 | Book-Crossing |
|---|---|---|---|
| #Sequence | 71,567 | 31,668 | 278,858 |
| #Items | 10,681 | 38,048 | 271,379 |
| #Interactions | 10,000,054 | 1,561,406 | 1,149,780 |

### B.1. Datasets

Table 2 presents the statistical overview of all datasets, specifying the sequential dimensions (user counts), item quantities, and interaction records. A concise description of each dataset follows:

- **Movielens-10M**. This is a classic movie rating dataset released by the GroupLens team, containing approximately 10 million user-movie rating records (1-5 stars), covering around 71,000 users and 10,000 movies.

- **Yelp2018**. This dataset originates from the business and user interaction data released by the Yelp platform in 2018. It is primarily used for research in recommendation systems, social network analysis, and text mining, containing user ratings and reviews for businesses (such as restaurants and bars) as well as social relationship information.

- **Book-Crossing**. The Book-Crossing dataset is a publicly available book rating dataset collected and organized by Cai-Nicolas Ziegler in 2004 from the BookCrossing.com community. It contains users' explicit ratings (1-10 points) and implicit feedback for books, primarily used for research in recommendation systems, collaborative filtering algorithms, and user behavior analysis

It is worth noting that we retain only users with at least 20 interactions in the ML-10M and Book-Crossing datasets, and users with at least 40 interactions in the Yelp2018 dataset. For ML-10M and Yelp2018, interactions with ratings greater than 4 are considered positive samples, while for Book-Crossing, interactions with ratings above 7 are treated as positive samples. We will process the above dataset to construct four common types of OOD.

- **Popularity shift**. We randomly sample 20% of interactions to create the out-of-distribution test set, maintaining a balanced distribution of item popularity. The rest of the data is divided into training, validation, and in-distribution (IID) test sets with a 7:1:2 ratio. This distribution shift is applied to the Yelp2018 datasets.

- **Temporal shift**. To simulate a realistic distribution shift, we organize the dataset by reverse chronological order and extract the latest 20% of each user's interactions as the OOD test set. The remaining portion is partitioned into training (70%), validation (10%), and IID test (20%) sets. This methodology is implemented on the Movielens-10M dataset.

- **Exposure shift**. Following the same data processing approach, we first sort the data chronologically and designate the most recent 20% of interactions as the OOD test set. The remaining data is split into training, validation, and IID test sets in a 7:1:2 ratio. It should be noted that exposure shift requires the test set to consist of fully exposed data, which is difficult to obtain in practice. Therefore, we directly adopt the approach from prior works (Bonner & Vasile, 2018; Saito, 2020; Wei et al., 2021), using a matrix factorization scheme to impute missing ratings in the test set, simulating a fully exposed scenario.

- **Mixed shift**. We construct a mixed-shift test set on Yelp2018 by combining two *in-domain* OOD subsets to simulate multiple realistic shifts. Specifically, we build (i) a *popularity-shift* OOD test split and (ii) a *temporal-shift* OOD test split, where test interactions come from a later (future) time window that does not overlap with the training period. We then form the mixed-shift test distribution by sampling 80% interactions from (i) and 20% from (ii), yielding a test set that simultaneously exhibits popularity and temporal shifts without introducing any out-of-dataset noise.

### B.2. Baselines

We evaluate CausalDPO against both conventional recommendation approaches and LLM-based benchmarks to demonstrate our method's effectiveness. Specifically, among traditional recommendation methods, we include the following models

- SASRec (Kang & McAuley, 2018) is a commonly adopted baseline framework that utilizes sequential modeling enhanced by self-attention mechanisms.

- BigRec (Bao et al., 2025) acts as a prompt-optimized foundation model for sequence-aware recommendations.

- RW (Jiang et al., 2024) provides a solution to item-side fairness issues in LLM-based recommendation caused by the combined effects of user historical behavior biases and inherent LLM semantic biases.

- $D^3$ (Bao et al., 2024). The proposed method pioneers a dual-debiasing mechanism to resolve both score amplification bias and recommendation homogeneity in LLM-based recommendations, caused by incompatible decoding mechanisms.

- DMPO (Bai et al., 2024) employs a pairwise ranking loss to enhance LLM-based recommendation performance by simultaneously maximizing the probability of positive samples and minimizing the probabilities of multiple negative samples.

- SDPO (Chen et al., 2024) enhances recommendation accuracy and ranking capability by integrating the Plackett-Luce ranking model with softmax sampling (i.e., constructing multiple negative samples), enabling automatic positive-negative differentiation and hard negative mining during fine-tuning.

- RosePO (Liao et al., 2024) improves generative recommendation accuracy while reducing semantic hallucination and popularity bias by explicitly modeling comparative user preference relationships (e.g., chosen vs. rejected items) and incorporating personalized smoothing optimization mechanisms.

- SPRec (Gao et al., 2024). The core approach of SPRec combines supervised fine-tuning and direct preference optimization through a self-play iterative framework, dynamically generating adversarial negative samples (predictions from previous iterations) to mitigate LLM-based recommendation bias.

### B.3. Implementation Details

We employ Llama-3.1-8B as the base language model. To ensure experimental fairness, all baseline methods and CausalDPO utilize the same supervised fine-tuning dataset. In implementation, each language model-based method undergoes 3 complete training epochs. It is noteworthy that in the main experimental results, all DPO experiments use only a single negative sample, where CausalDPO follows the approach in (Chen et al., 2024) by generating negative samples through random sampling. SPRec (Gao et al., 2024) employs single-iteration SFT model outputs to build its negative samples. The traditional SASRec model uses the same training and validation sets as other LLM-based methods, with the embedding dimension fixed at 32 and the dropout rate set to 0.1. All experiments were conducted on a computing node equipped with 8 NVIDIA A800 GPUs (80GB VRAM per GPU). Detailed hyperparameter configurations can be found in the project code repository.

*Table 3.* The performance (%) comparison between the baselines and CausalDPO on the two datasets with three **IID distributions**. We highlight the methods with the **best** and second-best performances.

| Methods | ML-10M | | | | Book-Crossing | | | |
|---|---|---|---|---|---|---|---|---|
| | H@10 | N@10 | H@20 | N@20 | H@10 | N@10 | H@20 | N@20 |
| SASRec | 0.47 | 0.17 | 0.75 | 0.27 | 1.70 | 0.90 | 3.12 | 1.02 |
| BIGRec | 0.66 | 0.31 | 1.00 | 0.40 | 1.64 | 0.73 | 2.19 | 0.87 |
| RW | 0.77 | 0.45 | 1.03 | 0.45 | 1.52 | 0.81 | 1.89 | 0.95 |
| $D^3$ | 0.72 | 0.41 | 0.99 | 0.40 | 1.48 | 0.71 | 2.03 | 0.82 |
| DMPO | 0.63 | 0.33 | 0.97 | 0.42 | 1.61 | 0.79 | 2.25 | 0.91 |
| SDPO | 0.60 | 0.36 | 0.94 | 0.45 | 1.59 | 0.75 | 2.17 | 0.88 |
| RosePO | 0.58 | 0.29 | 0.89 | 0.38 | 1.60 | 0.77 | 2.00 | 0.91 |
| SPRec | 0.60 | 0.31 | 0.95 | 0.41 | 1.52 | 0.70 | 2.31 | 0.90 |
| Ours | **0.79** | **0.48** | **1.12** | **0.51** | **1.77** | **0.99** | **3.28** | **1.18** |

*Table 4.* Model Performance (%) Comparison Across Two Distribution Shift Types. We highlight the methods with the **best** and second-best performances.

| Metric | SASRec | BIGRec | RW | $D^3$ | DMPO | SDPO | RosePO | SPRec | Ours |
|--------|--------|--------|------|------|------|------|--------|-------|------|
| H@10 | 0.48 | 0.49 | 0.25 | 0.45 | 0.50 | 0.51 | 0.39 | 0.41 | **0.53** |
| N@10 | 0.20 | 0.22 | 0.08 | 0.18 | 0.14 | 0.15 | 0.19 | 0.20 | **0.24** |
| H@20 | 0.72 | 0.76 | 0.51 | 0.68 | 0.75 | 0.76 | 0.70 | 0.75 | **1.51** |
| N@20 | 0.25 | 0.28 | 0.15 | 0.23 | 0.20 | 0.21 | 0.26 | 0.28 | **0.47** |

*Table 5.* Comparison of models with and without causal regularization on ML-10M and Book.

| Methods | ML-10M | | | | Book-Crossing | | | |
|---------|--------|--------|--------|--------|--------|--------|--------|--------|
|         | H@10 | N@10 | H@20 | N@20 | H@10 | N@10 | H@20 | N@20 |
| Dr.DPO | 0.17 | 0.20 | 0.55 | 0.18 | 0.39 | 0.19 | 0.50 | 0.26 |
| CPO | 0.41 | 0.25 | 0.81 | 0.31 | 0.34 | 0.20 | 0.54 | 0.24 |
| SimPO | 0.29 | 0.21 | 0.58 | 0.23 | 0.39 | 0.18 | 0.61 | 0.24 |
| DPO | 0.43 | 0.20 | 0.81 | 0.29 | 0.19 | 0.10 | 0.39 | 0.20 |
| SimPO w/ Causal Reg. | 0.36 | 0.27 | 0.64 | 0.34 | 0.45 | 0.03 | 0.70 | 0.30 |
| CPO w/ Causal Reg. | 0.52 | 0.31 | 0.95 | 0.39 | 0.47 | 0.33 | 0.73 | 0.32 |
| Dr.DPO w/ Causal Reg. | 0.58 | 0.33 | 0.98 | 0.41 | 0.49 | 0.35 | 0.75 | 0.34 |
| **CausalDPO** | **0.61** | **0.35** | **1.02** | **0.45** | **0.64** | **0.37** | **1.08** | **0.48** |

## B.4. Further Ablation Studies Analysis

Performance Comparison with Other DPO Variants. In Table 5, we further compare several DPO variants originally designed for other tasks. Dr.DRO(Wu et al., 2024a) integrates Distributionally Robust Optimization (DRO) theory(Rahimian & Mehrotra, 2022) to improve DPO's robustness against noisy samples. SimPO(Meng et al., 2024) adopts the average log-probability as an implicit reward and introduces a reward margin to better distinguish between preferred and non-preferred responses. CPO(Guo et al., 2024) incorporates a controllability mechanism to balance multiple preference objectives (e.g., accuracy and diversity), enabling more fine-grained alignment and response optimization.

As shown in Table 5, various existing DPO variants generally underperform compared to our proposed CausalDPO on out-of-distribution (OOD) datasets. The core reason for this performance gap lies in the difference in design objectives. From the outset, CausalDPO is explicitly designed to model and intervene on environment confounders in OOD recommendation scenarios, aiming to eliminate generalization errors caused by spurious correlations in preference learning. This enables the model to achieve robust generalization when facing complex distributional shifts. In contrast, other DPO variants are often tailored for specific goals, such as robustness to noise or enhanced controllability, but lack systematic modeling of environmental factors. As a result, they often struggle to maintain consistency and robustness across environments in OOD settings. It is also worth noting that the performance of these DPO variants varies across datasets. For example, on the MovieLens dataset, CPO achieves results close to standard DPO, while Dr.DPO and SimPO perform significantly worse. This discrepancy may stem from differences in the datasets themselves, such as noise levels, the complexity of user preference patterns, or the identifiability of latent environmental factors. These factors can lead to performance fluctuations

*Table 6.* Computational Complexity Comparison of Models

| Methods | DPO | CPO | SimPO | CausalDPO |
|---------|-----|-----|-------|-----------|
| Complexity | $\mathcal{O}(3B \cdot L^2 \cdot d)$ | $\mathcal{O}(3B \cdot L^2 \cdot d + BnLd)$ | $\mathcal{O}(2B \cdot L^2 \cdot d)$ | $\mathcal{O}(B \cdot L^2 \cdot d + B^2)$ |
| **seconds / epoch** | 2482s | 2160s | 1445s | 2971s |

*Table 7.* Computational Complexity Comparison of Models with Causal Invariant Regularization

| Methods | Dr.DPO w/ Causal | CPO w/ Causal | SimPO w/ Causal | CausalDPO |
|---|---|---|---|---|
| seconds / epoch | 2504s | 2659s | 2280s | 2971s |

*Table 8.* Ablation study results (%).

| Variant | Yelp2018 | | ML-10M | | Book-Crossing | |
|---|---|---|---|---|---|---|
| | H@10 | N@10 | H@10 | N@10 | H@10 | N@10 |
| w/o SFT | 1.28 | 0.54 | 0.52 | 0.26 | 0.25 | 0.12 |
| w/o CausalDPO | 1.47 | 0.69 | 0.55 | 0.28 | 0.52 | 0.30 |
| w/o Both | 1.32 | 0.37 | 0.46 | 0.22 | 0.27 | 0.12 |
| **Ours** | **1.53** | **0.77** | **0.58** | **0.32** | **0.64** | **0.37** |

when the models lack adequate modeling of environment-specific confounders. Moreover, we further integrated the proposed CausalDPO framework into various DPO variants, constructing CausalDPO implementations based on different backbones. For example, SimPO w/ Causal Reg. refers to the SimPO framework enhanced with causal invariance regularization. Experimental results show that SimPO w/ Causal Reg. achieves substantial performance improvements over the original SimPO on the ML-10M dataset across multiple evaluation metrics—for instance, achieving a 28.57% gain in NDCG@10. Similar improvements are observed across other DPO variants, demonstrating consistent and stable performance gains. These findings further validate the strong generality of our proposed causal modeling mechanism. CausalDPO serves as a modular enhancement that can be flexibly integrated into a wide range of existing DPO frameworks, effectively boosting generalization performance in out-of-distribution (OOD) recommendation scenarios regardless of the underlying architecture.

In addition, Table 7 compares the training time changes after incorporating causal regularization into various DPO variants. Taking SimPO as an example, its per-epoch training time on the Book-Crossing dataset increased from 1445 seconds (as shown in Table (6)) to 2280 seconds. While this represents a moderate increase, the significant performance improvement achieved in OOD generative recommendation tasks makes the additional cost justifiable.

**Time Complexity Analysis**. In addition, Table 6 compares the theoretical time complexity and empirical runtime per epoch of the original DPO, its variants, and CausalDPO. The theoretical time complexity of CausalDPO is $\mathcal{O}(\mathcal{B} \cdot L^2 \cdot d + \mathcal{B}^2)$, where $B$ is the batch size, $L$ is the number of input tokens, and $d$ is the embedding dimension. The $\mathcal{B}^2$ term mainly arises from the MMD computation or the pairwise environment consistency loss. In terms of complexity, CausalDPO introduces a slight increase over the original DPO. Empirically, we compare the per-epoch fine-tuning time on the Book-Crossing dataset: CausalDPO takes 2971 seconds, while DPO takes 2482 seconds, indicating a 19.70% increase in runtime. Nevertheless, CausalDPO achieves an average performance improvement of approximately 205.9% over DPO on this dataset, demonstrating a favorable trade-off between performance and computational cost. The additional overhead in CausalDPO mainly comes from its causal invariance regularization: the model first performs soft clustering to infer the latent environment of each sample, then enforces distributional consistency across environments using invariant learning constraints. Additionally, since SimPO does not rely on reference model alignment, it exhibits the shortest runtime.

**Different Backbone Model Sizes.** We further evaluate CAUSALDPO with backbones of different parameter scales to examine the robustness of our method across model capacities. Specifically, we adopt *Qwen2.5-7B-Instruct* and *Qwen3-8B-Instruct* as the backbone models. The results are reported in Table 9. Overall, CAUSALDPO consistently benefits from stronger backbones: upgrading from 7B to 8B yields a clear and uniform improvement on both Yelp2018 and ML-10M across all metrics. Moreover, CAUSALDPO remains competitive even with the 7B backbone, achieving comparable performance to the default setting and showing stable gains over strong baselines under OOD distributions. These observations indicate that the effectiveness of CAUSALDPO does not hinge on a specific model scale; instead, it generalizes well across different backbone capacities while scaling favorably with model size, suggesting good practicality for deployment under varying compute budgets.

*Table 9.* Under two **OOD distribution** settings on two datasets, we compare the performance (%) of CAUSALDPO with different model sizes against strong baselines. The **best** results are highlighted in **red**, and the second-best results are highlighted in blue.

| Methods | Yelp2018 | | | | ML-10M | | | |
|---|---|---|---|---|---|---|---|---|
| | **H@10** | **N@10** | **H@20** | **N@20** | **H@10** | **N@10** | **H@20** | **N@20** |
| SASRec | 1.01 | 0.43 | 1.51 | 0.61 | 0.50 | 0.02 | 0.84 | 0.30 |
| BIGRec | 0.77 | 0.50 | 1.28 | 0.62 | 0.53 | 0.29 | 0.76 | 0.35 |
| RW | 0.51 | 0.42 | 1.02 | 0.56 | 0.50 | 0.28 | 0.74 | 0.30 |
| D3 | 0.63 | 0.38 | 1.15 | 0.51 | 0.50 | 0.25 | 0.80 | 0.34 |
| **Ours** | 1.53 | 0.77 | 1.79 | 0.82 | 0.61 | 0.35 | 1.02 | 0.45 |
| *-Qwen2.5-7B-Instruct* | 1.50 | 0.76 | 1.77 | 0.80 | 0.60 | 0.35 | 1.02 | 0.44 |
| *-Qwen3-8b-Instruct* | 1.54 | 0.80 | 1.81 | 0.85 | 0.63 | 0.38 | 1.09 | 0.47 |

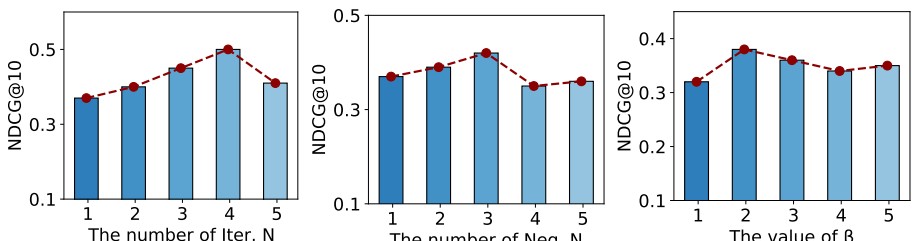

*Figure 3.* Evaluating the impact of model hyperparameter sets on recommendation performance.

## B.5. Hyperparameter Analysis

Figure 3 presents the impact of different hyperparameter settings on the performance of CausalDPO on the Book-Crossing dataset (measured by NDCG@10), with the following key observations: (1) The number of training iterations $N$ also plays a critical role. By adopting the iterative optimization strategy proposed in (Gao et al., 2024), we observe that the model achieves its best performance after four iterations, directly validating that iterative training can effectively enhance model performance. (2) The third plot in Figure 3 shows the impact of the number of negative samples. While using multiple negative samples generally boosts performance by improving contrastive learning, an excessive number increases training time and may dilute informative learning signals. (3) Finally, the value of $\beta$, which controls the relative weight between positive and negative pairs in the loss, also affects model performance. The best result is observed at $\beta$=2, indicating that a balanced contrastive signal is essential. Extremely low or high values may lead to suboptimal learning.

---

**Algorithm 1** Causal Direct Preference Optimization Fine-tuning Procedure

---

**Input:** Policy model $\pi_\theta$, reference model $\pi_{\text{ref}}$, dataset $\mathcal{D}$ with tuples $(x, y_w, y_l)$, batch size $\mathcal{B}$
**Hyperparameters:** temperature $\beta$, regularization weight $\lambda$
**Output:** Fine-tuned policy model parameters $\pi_\theta$

1: **for** each training iteration **do**
2:     Sample a mini-batch $\{(x, y_{w,i}, y_{l,i})\}_{i=1}^{\mathcal{B}}$ from $\mathcal{D}$
3:     Compute hidden states $h_i = \text{LLM}(x)$ and causal representations $z_i = g(h_i)$ for all $x$
4:     Apply DBSCAN to $\{z_i\}_{i=1}^{B}$ to obtain clusters and centers $\{c_k\}_{k=1}^{K}$
5:     Compute soft assignments $p_{ik}$, aggregated representations $\bar{x}^{(k)}$ according to Eq. 8 and Eq. 9
6:     Compute DPO loss $\mathcal{L}_{\text{DPO}}$ over the batch
7:     Compute MMD penalty MMD across pseudo-environments
8:     Compute total loss: $\mathcal{L}_{\text{CausalDPO}} = \mathcal{L}_{\text{DPO}} + \lambda \cdot \text{MMD}$
9:     Compute gradient $\nabla_\theta \mathcal{L}_{\text{total}}$ and update: $\theta \leftarrow \theta - \eta \cdot \nabla_\theta$
10: **end for**

---

