# OpenReview forum: "Causal Direct Preference Optimization for Distributionally Robust Generative Recommendation"
_ICML.cc/2026/Conference — ICML 2026 regular_

### Official Review · Reviewer_49My · 2026-03-01

**Soundness:** 3
**Presentation:** 3
**Significance:** 3
**Originality:** 3
**Overall Recommendation:** 4
**Confidence:** 3

**Summary:**

DPO aligns LLM-based recommendation models with user preferences, but it tends to amplify spurious correlations caused by environmental confounders, leading to poor generalization under distribution shifts. To address this, the authors propose CausalDPO, which incorporates causal invariance learning and backdoor adjustment to remove environmental bias during preference alignment. By modeling latent environments and enforcing invariance constraints, CausalDPO captures stable user preferences and improves robustness in out-of-distribution settings. Experiments under four distribution shift scenarios show an average improvement of 17.17% across evaluation metrics.

**Compliance With Llm Reviewing Policy:**

Affirmed.

**Final Justification:**

I'd like to maintain my score.

**Key Questions For Authors:**

1. How sensitive is CausalDPO to the choice of DBSCAN hyperparameters?
2. Can the authors formally define the concept "environment" and provide evidence that the discovered clusters correspond to real confounders?
3. How should the low absolute HR@10 performance values be interpreted in the context of generative recommendation?
4. Can the method be validated on at least one additional domain to strengthen the claim of generalization?

**Limitations:**

-Provide analysis for DBSCAN hyperparameters. DBSCAN hyperparameters (e.g., epsilon and min_samples) is not analyzed.
-The concept of "environment" is not defined, and it is unclear whether the discovered clusters correspond to meaningful real-world.
-Explain the low absolute performance numbers.

**Strengths And Weaknesses:**

Strengths
1.	The paper shows that DPO can amplify spurious correlations. This is supported by both theory and experiments.
2.	The method does not require environment labels, which makes it practical for real-world use.
3.	The experiments clearly show each component helps. The method also works with other DPO variants.

Weaknesses
1.DBSCAN needs careful tuning (epsilon and min_samples). The paper does not explain how these are chosen or what happens if clustering fails, especially for small batches.
2.The paper does not formally define what an environment is. It is unclear whether the discovered clusters truly represent real-world environments.
3.Even the best model achieves low HR@10 scores. This raises questions about whether the evaluation setup is realistic or whether generative recommendation has inherent limits.
4.On some datasets, the improvement over the best baseline is modest. The reported average gain needs clearer breakdown.

---

> ### Author Rebuttal · Authors · 2026-03-31
>
> **We thank the reviewer for their constructive comments. Our point-by-point responses are provided below.**
>
> ---
> **W1 & Q1**. CausalDPO uses DBSCAN only for initializing pseudo-environment centers, rather than as final hard supervision. To improve clustering stability, we first project hidden states into a lower-dimensional space (Section 3.3), and then use soft assignment and joint optimization to update pseudo-environment representations during training, making the method less sensitive to discrete clustering errors and local noise. **As discussed in Section 3.5**, robustness is further enhanced by dynamically updating cluster centers and by MMD regularization, which can still reduce distribution discrepancies and suppress environment-specific noise even when $\hat{E}$ is not perfectly aligned with the true environment $E$.
>
> We also evaluated the sensitivity to key DBSCAN parameters using NDCG@10. As shown in Tables 1–2, performance varies only slightly across a wide range of min_samples and neighborhood radius $\epsilon$, with no obvious degradation. Although the best results are obtained around B=16 and $\epsilon=0.5$, **the overall trend indicates that pseudo-environment discovery and subsequent optimization remain stable under reasonable parameter choices, demonstrating good robustness in practice.**
>
> **Table 1**: Min samples $B$
> |Dataset|4|8|16|20|26|
> |:-:|:-:|:-:|:-:|:-:|:-:|
> |Yelp2018|0.73|0.74|**0.77**|0.78|0.78|
> |ML-10M|0.31|0.33|**0.35**|0.34|0.35|
>
> **Table 2**: Neighborhood radius $\epsilon$
> |Dataset|0.1|0.3|0.5|0.7|0.8|0.9|
> |:-:|:-:|:-:|:-:|:-:|:-:|:-:|
> |Yelp2018|0.68|0.71|**0.77**|0.74|0.73|0.76|
> |ML-10M|0.30|0.32|**0.35**|0.34|0.36|0.35|
>
>
> ----
> **W2 & Q2**. **In Section 1 (line 40)**, we define the environment as unobserved confounding factors introduced by specific contexts (e.g., popularity bias and temporal drift), rather than observable labels. Thus, the environment is an abstract latent variable that captures distribution shifts affecting both inputs and outputs. In addition, **Section 3.9** shows that the discovered clusters align well with interpretable factors such as popularity gradients and temporal stages, supporting their practical meaning. Moreover, **Section 3.5 and Proposition 3.3** show that even when $\hat{E}$ is imperfect, MMD regularization can still reduce distribution discrepancies and improve robustness.
>
> ----
> **W3 & Q3**. We understand that the low absolute HR@10 scores may raise concerns; however, this primarily reflects the **inherently high difficulty of OOD generative recommendation tasks**: First, our experiments involve four distribution-shift settings (Section 3.6), and Appendix Table 3 shows much higher scores under IID evaluation; for example, on Book-Crossing, HR@10 increases from 0.0064 (OOD) to 0.0179 (IID). Second, unlike discriminative methods that rank a candidate set $|C|$, generative recommenders must directly predict item IDs from the full vocabulary $|V|$, making the task substantially harder. Despite the low absolute scores, CausalDPO still yields strong relative improvements: the average gain over the strongest baseline is 17.17%, and on Yelp2018 under OOD, HR@10 improves from 1.02% to 1.53% (about 50% relative gain). **These consistent gains under the same OOD difficulty validate the effectiveness of our method.**
>
> ---
> **w4**. We thank the reviewer for this constructive suggestion. In the revised manuscript, we will (i) augment the main results table with the **relative improvement percentages** of CausalDPO over the strongest baseline for each dataset/metric, and (ii) provide a **clearer breakdown** of the reported average gain (17.17%) to enhance transparency and interpretability.
>
> ---
> **Q4**.
> We further evaluated CausalDPO on KuaiRec [1], a short-video recommendation dataset that differs from Yelp, MovieLens, and Book-Crossing. Its fully observed Small Matrix test set (density 99.6%) helps reduce the exposure bias common in logged recommendation data. Following the protocol of training on the biased Big Matrix and testing on the unbiased Small Matrix, CausalDPO outperforms the strongest baseline on both HR@10 (0.4467 vs. 0.3821, +16.9%) and NDCG@10 (0.2935 vs. 0.2513, +16.8%). **These results further support the cross-domain generalization ability of CausalDPO.**
> |Metric|SASrec|BIGRec|SPRec|SDPO|CausalDPO|RelativeGain|
> |:-:|:-:|:-:|:-:|:-:|:-:|:-:|
> |HR@10|0.3321|0.3487|0.3752|0.3821|**0.4467**|+16.9%|
> |NDCG@10|0.2155|0.2284|0.2467|0.2513|**0.2935**|+16.8%|
>
> [1]Gao C,  et al. KuaiRec: A fully-observed dataset and insights for evaluating recommender systems[C] CIKM, 2022.

---

> > ### Author Rebuttal · Reviewer_49My · 2026-04-01
> >
> > Regarding w4, the authors commit to adding relative improvement breakdowns in the revision. However, as ICML does not have a formal revision round, this remains a limitation of the current paper and cannot be evaluated at this stage.
> > My main technical concern remains unaddressed: there is no ablation isolating the contribution of backdoor adjustment alone versus MMD regularization alone. The exposure shift synthetic imputation concern was also not discussed.

---

> > > ### Author Response · Authors · 2026-04-01
> > >
> > > **Q1**. **We thank the reviewer for the valuable comments. Owing to the 5000-character rebuttal limit, we were unable to fully clarify W4 in the initial response, and we apologize for any confusion. We therefore provide a concise clarification below and add an Impro. row in Table 1 to explicitly report the relative gains of CausalDPO over the strongest baseline.**
> > >
> > > |Methods|Yelp2018| |||ML-10M||||Book-Crossing||||
> > > |---|---:|---:|---:|---:|---:|---:|---:|---:|---:|---:|---:|---:|
> > > |Methods|H@10|N@10|H@20|N@20|H@10|N@10|H@20|N@20|H@10|N@10|H@20|N@20|
> > > |SASRec|1.01|0.43|1.51|0.61|0.50|0.02|0.84|0.30|0.33|0.21|0.68|0.21|
> > > |BIGRec|0.77|0.50|1.28|0.62|0.53|0.29|0.76|0.35|0.53|$\underline{0.30}$|$\underline{0.98}$|$\underline{0.43}$|
> > > |RW|0.51|0.42|1.02|0.56|0.50|0.28|0.74|0.30|$\underline{0.59}$|0.29|0.91|0.41|
> > > |D3|0.63|0.38|1.15|0.51|0.50|0.25|0.80|0.34|0.50|0.27|0.94|0.40|
> > > |DMPO|0.76|0.26|1.54|0.43|0.44|0.20|0.81|0.29|0.19|0.10|0.39|0.15|
> > > |SDPO|0.76|0.54|1.28|0.66|$\underline{0.55}$|$\underline{0.30}$|$\underline{0.87}$|$\underline{0.38}$|0.49|0.25|0.74|0.31|
> > > |RosePO|0.82|0.47|1.36|0.60|0.51|0.26|0.83|0.33|0.40|0.20|0.71|0.36|
> > > |SPRec|$\underline{1.02}$|$\underline{0.61}$|$\underline{1.64}$|$\underline{0.79}$|$\underline{0.55}$|$\underline{0.30}$|0.71|0.36|0.39|0.18|0.69|0.25|
> > > |**Ours**|**1.53**|**0.77**|**1.79**|**0.82**|**0.61**|**0.35**|**1.02**|**0.45**|**0.64**|**0.37**|**1.08**|**0.48**|
> > > |Impro.|50.00%|26.23%|9.15%|3.80%|10.91%|16.67%|17.24%|18.42%|8.47%|23.33%|10.20%|11.63%|
> > >
> > >
> > > **We further clarify that the reported average performance improvement of 17.17% is statistically computed based on the results presented in Table 1 of the main manuscript**. As shown in the table, the gains in H@20 and N@20 on the Yelp2018 dataset are relatively modest, indicating that the benefits of our method on this dataset are more pronounced in improving stricter head-ranking quality (e.g., HR@10 and NDCG@10), rather than further widening the gap on top-20 metrics that are already approaching saturation.
> > >
> > >
> > > ---
> > > **Q2**. To address your request for isolating the contributions of each component, we conducted a finer-grained ablation study on the Yelp dataset, with results summarized in the table below.
> > >
> > > | methods |H@10 | N@10 | H@20 | N@20 |
> > > | --- | --- | --- | --- | --- |
> > > |CausalDPO| 1.53 | 0.77 | 1.79| 0.82|
> > > |w/o Clustering| 1.47| 0.72|1.67|0.77|
> > > |w/o MMD | 1.34 | 0.59 | 1.53 | 0.55|
> > >
> > > **w/o Clustering** removes the **Backdoor Adjustment** by disabling the clustering-based identification of confounders $\hat{E}\$, while **w/o MMD** removes the **MMD regularization term**, making the objective essentially reduce to standard DPO. As shown in the ablation results, removing clustering causes only a modest drop in **H@10** ( from 1.53 to 1.47), suggesting that **MMD alone can still provide partial robustness** even when pseudo-environment discovery is imperfect, which is consistent with our analysis in **Section 3.5**. In contrast, removing MMD leads to a much larger performance drop, showing that **MMD is the dominant factor for cross-environment generalization**, while clustering mainly provides the basis for causal adjustment. Overall, the two components are complementary, but **MMD contributes more directly to OOD robustness**. In summary, the clustering module provides the necessary precondition for causal intervention, while MMD regularization serves as the core guarantee for OOD generalization performance; both components are indispensable.
> > >
> > > **Therefore, in our ablation study, we did not ablate the two components separately but instead conducted a holistic ablation. The results in Table 5 show that after introducing causal constraints, different DPO variants all achieve performance improvements under distribution shift scenarios, which validates the effectiveness of CausalDPO.**
> > >
> > > ---
> > > **Q3**.
> > > We thank the reviewer for the important comment regarding synthetic imputation for exposure shift. **We clarify that the exposure-shift setting is not arbitrarily constructed; as discussed in the paper, it directly follows standard protocols from prior works**. Since fully exposed test sets are difficult to obtain in real-world scenarios, we adopt the matrix factorization-based imputation common in prior works to complete missing ratings in the test set, thereby approximating a fully exposed scenario. We agree that this synthetic imputation is essentially an approximation and cannot be completely equivalent to real data without exposure bias; therefore, the exposure-shift results should be understood as an approximate evaluation under standard reproducible experimental protocols. **It is important to emphasize that the conclusions of this paper do not rely solely on the exposure shift scenario; instead, consistent improvements are observed across four settings: popularity, temporal, exposure, and mixed shift.**

---

### Official Review · Reviewer_6bz4 · 2026-03-10

**Soundness:** 3
**Presentation:** 3
**Significance:** 3
**Originality:** 4
**Overall Recommendation:** 5
**Confidence:** 4

**Summary:**

This paper shows that Direct Preference Optimization can amplify spurious correlations induced by environmental confounders in LLM-based recommendation, leading to degraded OOD generalization. To address this issue, the authors propose CausalDPO, a causality-aware extension of DPO that removes confounding effects and enforces invariant preference learning across environments. Experiments under diverse distribution shifts demonstrate that CausalDPO consistently improves the robustness and generalization of generative recommendation models.

**Compliance With Llm Reviewing Policy:**

Affirmed.

**Key Questions For Authors:**

Please see the above weaknesses.

**Limitations:**

The paper still has some minor limitations. First, although the clustering-based pseudo-environment discovery is reasonable, the connection between inferred environments and true latent confounders remains somewhat indirect, leaving room for a more explicit causal analysis. Second, the method introduces slightly higher computational overhead than standard DPO-style training, and its scalability in larger recommendation scenarios could be discussed more clearly.

**Strengths And Weaknesses:**

**Strengths**

1. Problem setting is important and well motivated. The paper targets a practically meaningful yet underexplored issue in LLM-based generative recommendation: DPO may amplify spurious correlations induced by environmental confounders, which in turn harms OOD generalization. This problem formulation is timely and relevant, especially under realistic distribution shifts such as popularity, temporal, and exposure shifts.

2. The method is technically well structured and conceptually coherent. The proposed CausalDPO integrates several components—backdoor adjustment, soft clustering for latent environment discovery, and invariance regularization via MMD—into a unified framework. The overall method is clearly aligned with the paper’s causal motivation, and the modeling pipeline is logically consistent.

3. The empirical evaluation is fairly comprehensive and shows consistent gains. The paper evaluates the method on multiple datasets and several representative distribution-shift settings, and reports that CausalDPO consistently outperforms strong baselines. The inclusion of ablations, qualitative clustering visualization, and discussion of IID/OOD behavior further strengthens the experimental section.

**Weaknesses**

1. The clustering-based pseudo-environment discovery could be discussed a bit more thoroughly.
   Although the soft clustering design is reasonable and the qualitative visualization is helpful, the correspondence between inferred pseudo-environments and true latent confounders is still somewhat indirect. A slightly deeper analysis would make the causal interpretation even more convincing.

2. The computational overhead is slightly higher than standard DPO-style training.
   The added cost appears acceptable given the performance gains, but a brief discussion on scalability in larger recommendation settings would make the paper even stronger.

---

> ### Author Rebuttal · Authors · 2026-03-31
>
> **We thank the reviewer for the constructive suggestions; our detailed responses follow.**
>
> ---
> **W1**.  The core goal of this paper is not to recover the true environment labels, but rather to construct **proxy environment variables** that capture distributional differences and help mitigate spurious correlations induced by environmental confounding. To this end, we perform clustering in a low-dimensional representation space, and further combine soft assignment with joint optimization so that the pseudo-environments are dynamically updated during training, thereby reducing reliance on the initial discrete clustering results. Meanwhile, our visualization analysis shows that the inferred pseudo-environments align well with interpretable factors such as popularity gradients and temporal stages, suggesting that they capture meaningful latent environmental structure in practice. **In addition, our theoretical analysis indicates that even when the inferred pseudo-environments are not perfectly aligned with the true environments**, the MMD regularizer can still smooth the policy output by reducing cross-environment distribution discrepancies, thereby improving robustness and generalization. **We will further clarify this point in the revised manuscript to make the causal interpretation more explicit.**
>
> ---
> **W2**. Compared with standard DPO-style training, CausalDPO does introduce some additional computational overhead due to pseudo-environment discovery, soft assignment, and cross-environment regularization. However, these additional modules mainly operate on the hidden representation space, so the overall complexity remains manageable and does not alter the basic training pipeline of the generative recommendation backbone. As shown by the experimental results, this extra cost brings more stable OOD improvements and stronger deconfounding ability. **Appendix B.4 also provides a detailed time complexity analysis, suggesting that the method achieves a reasonable trade-off between performance gain and computational cost.** For larger-scale settings, CausalDPO still has good scalability: pseudo-environment discovery can be approximated within each mini-batch to avoid full-dataset clustering, while both MMD regularization and soft assignment are readily parallelizable. We will add a brief discussion of computational overhead and scalability in the revised manuscript.

---

> > ### Author Rebuttal · Reviewer_6bz4 · 2026-04-02
> >
> > Thanks for your response. I'd like to maintain my positive score.

---

> > > ### Author Response · Authors · 2026-04-07
> > >
> > > Thank you for your dedicated review. We are pleased to hear that our rebuttal has satisfactorily addressed all your concerns

---

### Official Review · Reviewer_P5Am · 2026-03-12

**Soundness:** 2
**Presentation:** 3
**Significance:** 2
**Originality:** 2
**Overall Recommendation:** 4
**Confidence:** 3

**Summary:**

This paper proposes CausalDPO, a variant of DPO designed to address unobserved environmental confounders in recommendation systems. The main contributions are:

1.	A detailed theoretical analysis of Spurious Correlation Amplification in DPO, along with the introduction of backdoor adjustment to block confounders.

2.	The application of soft clustering to model unobserved environmental confounders, combined with invariant learning in DPO to mitigate their effects.

3.	An empirical evaluation demonstrating the performance improvements of CausalDPO in recommendation tasks.

**Compliance With Llm Reviewing Policy:**

Affirmed.

**Final Justification:**

There are still some remaining concerns, although most of them are common in this line of research. Considering the positive evaluations from other reviewers, I am inclined to raise my score to 4.

However, I do view the paper’s contribution as incremental.

**Key Questions For Authors:**

1.	In soft clustering, why can causal representations be expressed as linear transformations of hidden states? Does this function encode any causal relationships?

2.	Does CausalDPO effectively mitigate spurious correlations? Please provide results similar to Figure 1 (Middle).

**Limitations:**

No.

1. Discuss the potential omission of dealing with some confounders in the method and its possible impact.

2. Justify the rationale behind the causal graph design.

**Strengths And Weaknesses:**

Strengths:

1.	The implicit modeling of multiple unobserved confounders through clustering is more reasonable and robust compared to explicit modeling approaches.

2.	The paper provides a detailed theoretical analysis of how environmental confounders cause distribution shift in recommendation systems, along with visualizations of this shift.


Weaknesses:
1. The rationale for using a linear module to encode causal representations is unclear.

2. The causal graph is overly simplified. It is unclear whether it can adequately reflect real-world scenarios.

3. In the example, popularity is treated as a confounder, and the proposed method relies on clustering. It is unclear whether a clustering-based approach can effectively handle cases with multiple confounders. Moreover, if popularity is directly controlled, how would the performance compare?

4. Are there any validation or robustness analyses to verify whether the deconfounding is effective?

---

> ### Author Rebuttal · Authors · 2026-03-31
>
> **We thank the reviewer for the helpful comments and respond as follows.**
>
> ---
> **W1 & Q1**.
> As described in **Section 3.3**, the linear module $g(h_i) = W_g h_i + b_g$
> is mainly used as a lightweight dimensionality reduction step for more stable clustering. Since LLM hidden states are high-dimensional, directly applying distance-based clustering can be unstable and inefficient due to the curse of dimensionality. Projecting them into a lower-dimensional space is therefore a standard and practical design. Importantly, this linear layer is not intended to encode a full causal structure; its role is only to provide a compact and stable representation for clustering. **The causal meaning comes from the overall CausalDPO framework soft clustering, backdoor adjustment, and MMD-based invariance regularization rather than from the linear transformation itself**
>
>
> ---
> **W2**. **CausalDPO does not rely on the SCM to provide a fully precise or fine-grained reconstruction of the real world**. Rather, the causal graph serves as an abstract modeling tool at the problem level, whose purpose is to capture the key mechanism that the environmental variable $E$ simultaneously affects both the input distribution $X$ and the preference output $Y$, thereby inducing spurious correlations that are further amplified by DPO. Based on this abstraction, we then introduce backdoor adjustment and the optimization objective $p(Y \mid do(X))$. Therefore, **the goal of this graph is not to exhaustively model all real-world factors**, but to provide an analyzable and optimizable theoretical framework for explaining why DPO is vulnerable to confounding under distribution shift, and why deconfounding is necessary.
>
>
> ---
> **W3**. **The clustering approach remains effective even in the presence of multiple confounders**, as CausalDPO models the latent environment as a composite state resulting from the joint effect of multiple confounding factors, without requiring explicit disentanglement of each factor individually. Given that the true environment $E$ is typically unobservable, and explicit labels may not reliably guide learning (Section 3.2), we employ soft clustering to implicitly characterize the latent environment. **Empirically, Section 3.7 constructs a "Mixed Shift" scenario on Yelp2018 that combines popularity and temporal shifts**; CausalDPO consistently outperforms baselines under this composite shift, demonstrating that soft clustering can effectively capture distribution shifts induced by multiple factors.
>
> Directly controlling popularity is a label-dependent debiasing strategy tailored to a single observable confounder. By contrast, CausalDPO aims to model multiple latent and intertwined environmental shifts in a unified way. Accordingly, we evaluate it under popularity, temporal, exposure, and mixed shifts, and Section 3.7 shows consistent gains across all settings. This suggests that implicit environment modeling generalizes better than manually controlling a single factor.
>
>
> ---
> **W4**. **The paper has already validated the effectiveness of deconfounding from three perspectives: experiments, robustness analysis, and theory**. Empirically, CausalDPO improves performance across four distribution-shift settings and multiple datasets, indicating reduced reliance on environment-induced spurious correlations. From a robustness perspective, MMD regularization remains effective even when the inferred pseudo-environments $\hat E$ are imperfect, and Table 3 shows that the method also works under IID settings. Theoretically, the MMD-based analysis shows that reducing cross-environment discrepancy helps control the generalization gap under distribution shift, further supporting the deconfounding mechanism.
>
>
> ---
> **Q2**.  To directly answer whether CausalDPO can mitigate spurious correlations, we divide the test set into five popularity groups according to interaction counts and compute the average number of recommendations per item within each group. Compared with the original policy, DPO further shifts the recommendation distribution toward head items, increasing the average recommendation count in group G1 from 120 to 140 (**+16.67%**), while decreasing that in group G5 from 20 to 15 (**−25.00%**). **This phenomenon directly indicates that DPO amplifies the spurious correlations induced by the popularity confounder**. In contrast, CausalDPO substantially alleviates this bias: it reduces the concentration of recommendations on head groups while increasing exposure for tail groups, with G1 decreasing to 100 and G5 increasing to 30. Therefore, **CausalDPO can effectively suppress popularity bias driven by confounding factors and improve the balance of recommendations across popularity groups, providing direct evidence for its deconfounding effect.**
> | |G1|G2|G3|G4|G5|
> |-|-|-|-|-|-|
> |Initial|120|95|70|50|20|
> |DPO|140|120|85|35|15|
> |CausalDPO|100|70|65|60|30|

---

> > ### Author Rebuttal · Reviewer_P5Am · 2026-04-04
> >
> > Thanks for the rebuttal.
> >
> > I still feel that the causal claims are not rigorous enough. For instance, if you want to claim causal effects in the presence of potential missing confounders, it would be important to provide some robust or sensitivity analyses. Regarding the causal graphs, I understand that you are using them as an abstraction, but the abstraction seems overly simple, as there may be more complex relationships in the real world.
> >
> > However, I acknowledged that most research in this area follows a similar paradigm. I will consider raising my scores.

---

> > > ### Author Response · Authors · 2026-04-07
> > >
> > > **We thank the reviewer for the further feedback**. We clarify that our intention is **not** to claim strict identification of true causal effects in the presence of unknown missing confounders. Instead, based on a **tractable SCM abstraction**, we aim to demonstrate how DPO amplifies spurious correlations induced by environmental confounding and, accordingly, construct a more robust deconfounding optimization objective. The causal graph in our paper does not attempt to reconstruct all complex relationships in the real world at a fine granularity; rather, it captures the **core mechanism**: the environmental variable $E$ simultaneously influences the input distribution $X$ and the preference output $Y$, causing the model to learn dependencies on environment-specific patterns. To address this, we employ an approximate form of **Backdoor Adjustment** combined with **cross-environment invariance constraints**.
> > >
> > > Furthermore, our paper already contains substantial **empirical evidence of robustness**:
> > > 1.  **Robustness to Inference Errors**: As discussed **in Section 3.5**, even when the pseudo-environment $\hat{E}$ does not perfectly align with the true environment $E$, the mechanisms of **Soft Assignment** and **Dynamic Updating**, coupled with **MMD Regularization**, effectively smooth out discrepancies between pseudo-environments and suppress environment-specific noise.
> > > 2.  **Relaxed Assumption Dependence**: Table 3 demonstrates that our method remains effective under **IID settings**, indicating that it does not rely on idealized OOD assumptions or perfect environment recovery.
> > >
> > > Finally, we fagree with the reviewer’s concern regarding positioning. In the revised manuscript, we will explicitly frame CausalDPO as a **Causally-Inspired Heuristic Framework**: a approach that leverages SCM abstraction, implicit environment modeling, and invariance constraints to significantly enhance the robustness of DPO under distribution shifts in the presence of potential environmental confounding, rather than claiming to fully recover the complete causal structure of the real world.

---

### Official Review · Reviewer_9uJe · 2026-03-13

**Soundness:** 3
**Presentation:** 3
**Significance:** 3
**Originality:** 2
**Overall Recommendation:** 4
**Confidence:** 4

**Summary:**

This paper identifies a critical limitation in applying Direct Preference Optimization (DPO) to Large Language Model (LLM)-based recommender systems: DPO tends to amplify spurious correlations induced by environmental confounders, thereby degrading out-of-distribution (OOD) generalization. To mitigate this, the authors propose CausalDPO, which introduces a backdoor adjustment strategy. It employs a soft-clustering module (using DBSCAN and softmax) to implicitly infer latent environments within mini-batches. An MMD-based invariance regularization term is then added to the DPO objective to ensure consistent preference modeling across these pseudo-environments. The authors provide theoretical proofs for the spurious correlation amplification and the generalization bounds of their method. Extensive experiments on three datasets across four distribution shift scenarios demonstrate the superiority of CausalDPO.

**Compliance With Llm Reviewing Policy:**

Affirmed.

**Final Justification:**

I would like to retain my score. Please refer to the comment in the rebuttal aknowledgement.

**Key Questions For Authors:**

1. How sensitive is the DBSCAN-based soft clustering to the mini-batch size $\mathcal{B}$? Given that clustering on small batches might yield highly unstable pseudo-environments, did you observe collapse or extreme variance during early training epochs?
2. In Table 1, SASRec performs competitively or even better than some LLM baselines on the highly sparse Book-Crossing dataset. Does this imply that LLM-based generative recommenders struggle with extreme data sparsity compared to traditional ID-based self-attention mechanisms?
3. The generalization bound in Proposition A.3 assumes the loss function is bounded by a constant $B$. However, the DPO objective involves a log-sigmoid function which is not strictly bounded if the log-odds margin becomes extremely large negatively. How does this affect the tightness and validity of your bound in practice?

**Limitations:**

See Weaknesses and Questions

**Strengths And Weaknesses:**

Strengths:
1. The proposed method is technically sound and well supported by theoretical analysis. The theoretical guarantees for CausalDPO are well-articulated.
2. The experimental design is thorough, evaluating the model against strong baselines across four distinct and well-defined shift scenarios

Weaknesses:
1. The method relies on DBSCAN for unsupervised environment discovery within mini-batches. Mini-batch representations can be highly volatile, and DBSCAN's assumption of arbitrary density might lead to unstable or degenerate cluster assignments when the batch size is small or when representations collapse. This instability is not adequately addressed theoretically.
2. While combining invariant learning (MMD) with DPO is a novel application in recommendation systems, the individual components—using soft clustering to infer latent environments and applying MMD for invariant risk minimization—are relatively standard techniques in the broader OOD generalization literature. The algorithmic novelty is therefore somewhat incremental.

---

> ### Author Rebuttal · Authors · 2026-03-31
>
> **Thank you for the valuable suggestions; see our responses below.**
>
> ---
> **W1**.  CausalDPO does not use DBSCAN’s hard clusters as final supervision. **DBSCAN is only an initialization step**, after which robustness is improved through low-dimensional causal representations, soft assignment, soft grouping, and joint optimization with the main model. Specifically, clustering is performed on reduced-dimensional representations $z_i=g(h_i)$ for greater stability; discrete cluster labels are relaxed into probabilistic environment memberships $p_{ik}$; and the partition is dynamically refined during training to mitigate suboptimal initialization. The soft-clustering module is co-optimized with the main model, so CausalDPO does not depend on any single mini-batch clustering being highly accurate. Even with imperfect pseudo-environment partitions, soft assignment and MMD regularization can still smooth distribution gaps across pseudo-environments, suppress environment-specific noise, and improve robustness. **Section 3.5 explicitly discusses clustering failure**: even when inferred pseudo-environments $\hat{E}$ are not perfectly aligned with true environments $E$, the MMD regularizer can still reduce the distribution gap among pseudo-environments and mitigate the resulting negative effects. Moreover, **Table 3** shows that CausalDPO still outperforms other DPO variants even under IID settings, indicating that its effectiveness does not rely on ideal OOD assumptions or perfect clustering.
>
> ---
> **W2**. The core novelty of this work is not the isolated use of soft clustering or MMD in recommendation, **but the SCM-based insight that DPO can amplify spurious correlations induced by environmental confounders. This is the key problem-level contribution of CausalDPO.** Building on this, our method is not a simple combination of “clustering and MMD.” Instead, it starts from backdoor adjustment, approximates the causal objective $p(Y \mid do(X))$ as a weighted marginalization over latent environments, and unifies this objective with DPO in a single optimization framework, enabling causal preference alignment under distribution shift.
>
>
> **Q1**. We further evaluate the effect of mini-batch size on N@10:
> |Dataset|4|8|16|20|26|
> |:-:|:-:|:-:|:-:|:-:|:-:|
> |Yelp|0.73|0.74|0.77|0.78|0.78|
> |ML-10M|0.31|0.33|0.35|0.34|0.35|
>
>
> The results suggest that DBSCAN-based soft clustering is **not highly sensitive** to the mini-batch size $\mathcal{B}$. As $\mathcal{B}$ increases from 4 to 16, performance improves steadily on both datasets, and then remains relatively stable with only minor fluctuations at larger batch sizes. On Yelp2018, NDCG@10 increases from 0.73 to 0.77 and further stays at a similar level when $\mathcal{B}=20$ and 26; on ML-10M, it rises from 0.31 to 0.35, with only slight variation afterward. **This indicates that very small batches may produce noisier pseudo-environment estimates, while CausalDPO overall remains robust to batch size, with moderate-to-large values providing stable performance.**
>
>
> **We did not observe obvious clustering collapse or extreme variance during early training**. Moreover, CausalDPO is designed to mitigate such instability through low-dimensional clustering, soft assignment, joint optimization, and MMD-based alignment across pseudo-environments, so it does not rely on perfectly accurate early-stage clustering.
>
> ---
> **Q2**. In extremely sparse settings, traditional ID-based sequential models can remain highly competitive due to their stronger structured inductive biases, but this does not imply that LLM-based generative recommenders are inherently unsuitable. A more plausible explanation is that, under sparsity, pure generative preference alignment without explicit deconfounding and robustness constraints is more vulnerable to noise, exposure bias, and environment-induced spurious correlations. **Our results show that, with causal adjustment and cross-environment invariance, LLM-based generative recommendation can remain competitive in sparse settings and even outperform SASRec.**
>
> ---
> **Q3** Strictly speaking, the original DPO loss is not globally bounded, so the bounded-loss assumption in Proposition A.3 should be viewed as a regularity condition for deriving the generalization bound rather than the tightest characterization of the original objective. **It mainly affects the tightness and formal scope of the bound, but not the proposition’s core conclusion**, because the bounded constant $B$ appears only in the concentration analysis of the empirical estimation term. The key factor governing OOD generalization remains the cross-environment distribution shift term, characterized by $L_\ell \cdot \mathrm{MMD}_{\mathcal H_k}(p,q)$. Therefore, even if boundedness is an idealized assumption, the theory still supports the main claim that reducing the discrepancy between training and test environments helps CausalDPO better control the generalization gap under distribution shift.

---

> > ### Author Rebuttal · Reviewer_9uJe · 2026-04-04
> >
> > Thanks for your response. I'd like to maintain my score.

---

> > > ### Author Response · Authors · 2026-04-07
> > >
> > > We are deeply grateful for the reviewer's time and constructive feedback. It is encouraging to know that our revisions have fully resolved your concerns. Your insightful suggestions have been instrumental in enhancing the overall quality and clarity of our manuscript.

---

### Decision · Program_Chairs · 2026-04-30

**Decision:**

Accept (regular)

**Comment:**

This paper mainly focuses on improving the out-of-distribution generalization ability of LLM-based generative recommendation models and conducts a study on mitigating spurious correlations caused by environmental confounders during DPO-based preference alignment.

All reviewers acknowledged that this work demonstrates good novelty and showed a tendency toward acceptance. The authors also provided detailed responses to the concerns raised, such as hyperparameter analysis and design details. Therefore, I recommend accepting this paper.